# Copy-back viral genomes induce a cellular stress response that interferes with viral protein expression without affecting antiviral immunity

**Lavinia J. González Aparicio, Yanling Yang, Matthew Hackbart, Carolina B. López** [ID] *

Department of Molecular Microbiology and Center for Women Infectious Disease Research, Washington University School of Medicine in St. Louis, Missouri, United States of America

* clopezzalaquett@wustl.edu

**Data Availability Statement:** All relevant data are within the paper and its supporting information files. The sequencing raw data of Fig 6 is available in ncbi BioProject ID PRJNA1031189.

## Abstract

Antiviral responses are often accompanied by translation inhibition and formation of stress granules (SGs) in infected cells. However, the triggers for these processes and their role during infection remain subjects of active investigation. Copy-back viral genomes (cbVGs) are the primary inducers of the mitochondrial antiviral signaling (MAVS) pathway and antiviral immunity during Sendai virus (SeV) and respiratory syncytial virus (RSV) infections. The relationship between cbVGs and cellular stress during viral infections is unknown. Here, we show that SGs form during infections containing high levels of cbVGs, and not during infections with low levels of cbVGs. Moreover, using RNA fluorescent in situ hybridization to differentiate accumulation of standard viral genomes from cbVGs at a single-cell level during infection, we show that SGs form exclusively in cells that accumulate high levels of cbVGs. Protein kinase R (PKR) activation is increased during high cbVG infections and, as expected, is necessary for virus-induced SGs. However, SGs form independent of MAVS signaling, demonstrating that cbVGs induce antiviral immunity and SG formation through 2 independent mechanisms. Furthermore, we show that translation inhibition and SG formation do not affect the overall expression of interferon and interferon stimulated genes during infection, making the stress response dispensable for global antiviral immunity. Using live-cell imaging, we show that SG formation is highly dynamic and correlates with a drastic reduction of viral protein expression even in cells infected for several days. Through analysis of active protein translation at a single-cell level, we show that infected cells that form SGs show inhibition of protein translation. Together, our data reveal a new cbVG-driven mechanism of viral interference where cbVGs induce PKR-mediated translation inhibition and SG formation, leading to a reduction in viral protein expression without altering overall antiviral immunity.

## Introduction

Respiratory syncytial virus (RSV) and the parainfluenza viruses are endemic RNA viruses responsible for a large disease burden, especially involving children and older adults [1,2].

**Funding:** Financial support during preparation of this work was provided by the US National Institutes of Health National Institute of Allergy and Infections Diseases AI137062 and AI134862 (to CBL), and the Principles of Pulmonary Research Training Grant T32-007317 (to LGA and MH). The funders had no role in study design, data collection and analysis, decision to publish, or preparation of the manuscript.

**Competing interests:** The authors have declared that no competing interests exist.

**Abbreviations:** BSA, bovine serum albumin; cbVG, copy-back viral genome; CHX, cycloheximide; dKO, double KO; DMEM, Dulbecco's Modified Eagle's Medium; eIF2α, eukaryotic initiation factor 2 alpha; FBS, fetal bovine serum; FISH, fluorescence in situ hybridization; G3BP1, GTPase-activating protein-binding protein 1; hpi, hours postinfection; IFN, interferon; ISG, IFN-stimulated gene; KO, knockout; MAVS, mitochondrial antiviral signaling; MOI, multiplicity of infection; NP, nucleoprotein; nsVG, nonstandard viral genome; PKR, protein kinase R; PMY, puromycin; RIG-I, retinoic acid–inducible gene I; RLB, RNAseL-dependent body; RLR, RIG-I-like receptor; RSV, respiratory syncytial virus; SeV, Sendai virus; SG, stress granule; TIAR, TIA-1-related.

RNA viruses produce not only full-length standard viral genomes but also variants, hypermutated RNAs, and nonstandard viral genomes (nsVGs) that provide different functions and advantages to the virus [3,4]. nsVGs produced during RSV and parainfluenza virus infections are critical determinants of infection outcome in vitro and in vivo [5–7]. When produced early during infection, nsVGs significantly reduce virus spread and disease severity in mice and humans [5,7]. nsVGs impact the infection via stimulation of major signaling pathways that shape the cellular response to the infection. Identifying cellular pathways and molecular mechanisms by which nsVGs reduce virulence may lead to new strategies to prevent severe disease upon RNA virus infection.

One nsVG subpopulation, copy-back viral genomes (cbVGs), has critical roles in inducing the cellular antiviral immune response, controlling the rate of viral replication, and promoting the establishment of persistent infections [6–8]. Nonsegmented negative-sense RNA viruses generate cbVGs when the viral polymerase initiates replication at the promoter region, falls off the template, and then reattaches to the nascent strand [3]. The polymerase then uses the nascent strand as a template and continues replicating, copying back the already synthetized RNA (**S1A Fig**) [3]. The resulting RNA molecules contain highly structured immunostimulatory motifs and lack genes encoding viral proteins [7,9]. Although cbVGs can only replicate in the presence of a full-length standard genome that provides essential viral proteins, cbVGs are key interactors with the host and drive several cellular responses that determine the infection outcome. Notably, all the known effects of cbVGs on shaping the host response are dependent on the mitochondrial antiviral signaling (MAVS) pathway. cbVGs activate retinoic acid–inducible gene I (RIG-I)-like receptors (RLRs) leading to MAVS signaling, which then induces robust antiviral responses [9]. By activating the MAVS pathway, cbVGs stimulate the interferon (IFN) response that ultimately reduces virus spread and induces long-term protective immunity [6]. Additionally, cbVGs signal through MAVS to activate a cell survival mechanism that promotes the establishment of persistent infections in vitro [8]. Whether cbVGs can induce other cellular pathways that contribute to the outcome of the infection remains unknown.

In addition to the antiviral immune response, viral infections can induce cellular stress responses that lead to protein translation inhibition and the formation of stress granules (SGs) [10]. During most viral infections, the cellular stress response is initiated upon activation of the double-stranded RNA binding protein kinase R (PKR), which phosphorylates the eukaryotic initiation factor 2 alpha (eIF2α), leading to cap-dependent translation arrest, disassembly of polysomes, and formation of SGs. SGs are liquid phase-separated nonmembranous organelles composed mostly of untranslated mRNA and RNA binding proteins [11,12]. Activation of this cellular stress response during infection can lead to reduced viral protein expression [12] and has been proposed to mediate the antiviral immune response [13–17]. Accumulation of cbVGs during measles virus infection has been correlated with PKR activation, but whether cbVGs are the main triggers of PKR activation and SG formation is unknown [18].

The predicted overlapping antiviral roles of the PKR-driven cellular stress response and cbVGs led us to question if cbVGs are involved in SG formation during RSV and parainfluenza virus infections, and whether cbVG-mediated antiviral immunity depends on SG formation. Our data show that cbVGs are the primary inducers of canonical SGs during Sendai virus (SeV) and RSV infections through PKR activation, and that this induction is independent of the MAVS pathway. Contrary to previous reports, we found that MAVS does not localize to cbVG-induced SGs and that translation inhibition and SG formation are not required for overall induction of antiviral immunity. Instead, we show that cbVGs induce protein translation inhibition in SG-positive cells, resulting in reduced levels of viral proteins at a single-cell level without affecting the expression of antiviral proteins at a population level. Overall, these data demonstrate that cbVGs orchestrate the induction of cellular stress and antiviral

immunity independently, highlighting the importance of considering the presence of nsVGs when studying virus–host interactions. Importantly, our data uncover a new primary mechanism of interference by cbVGs via the induction of viral protein translational arrest.

## Results

### SGs form during RSV infection containing high levels of cbVGs

To assess whether cbVGs induced SG formation, we infected lung epithelial A549 cells with RSV stocks containing high or low levels of cbVGs (RSV cbVG-high and RSV cbVG-low, respectively). To achieve high and low cbVG accumulation in these stocks, the virus was grown at different multiplicity of infection (MOI), as virus expansion at high MOI promotes the accumulation of cbVG, while virus expansion at low MOI reduces the accumulation of cbVGs [19]. cbVG contents in the stocks were confirmed by PCR (**S1B Fig**). Because cbVGs potently induce the IFN response, we expect cbVG-high stocks to induce higher expression of *IL-29* than a cbVG-low stock [6]. As expected, *IL-29* mRNA levels were increased in cells infected with cbVG-high stocks (**S1C Fig**). Additionally, presence of cbVGs during infection is expected to correlate with reduced levels of virus replication in infected cells as compared to cbVG-low stocks due to the activity of IFNs [6]. Using *RSV G* mRNA transcripts as a proxy for virus replication, we confirmed that infection with RSV cbVG-high stocks resulted in reduced levels of *RSV G* mRNA as compared to infection with an RSV cbVG-low stocks (**S1B Fig**).

To visualize SG formation during RSV infections, cells were immunostained for the well-characterized SG associated protein Ras GTPase-activating protein-binding protein 1 (G3BP1), along with the RSV nucleoprotein (NP) to identify infected cells. Fluorescence imaging analysis showed SGs in infected cells during RSV cbVG-high infections, while they were rarely detected in RSV cbVG-low infections. SGs were observed as early as 12 hours postinfection (hpi) and were still present at 24 hpi (**Fig 1A**). The percent of SG-positive cells during RSV cbVG-high infection increased over time, and approximately 10% of infected cells were SG-positive at 24 hpi (**Fig 1B**). Of note, throughout our study, all SG-positive cells were positive for viral protein.

Although cbVG-containing viral particles can infect cells, they are not considered fully infectious as they can only replicate in cells coinfected with standard viral particles. Thus, infections based on MOI only account for the number of fully infectious particles in the inoculum. We expect that RSV cbVG-high infections, which contain both infectious standard particles and noninfectious cbVG particles, will contain a higher amount of total viral particles. To determine if the observed differences in SG formation were due to differences in total viral particles added in the inoculum, we infected cells with RSV cbVG-high and RSV cbVG-low at increasing MOIs and compared percent of SG-positive cells. Increasing the MOI of RSV cbVG-low infection did not increase the percent of SG-positive cells even when using 10 times more RSV cbVG-low than RSV cbVG-high (**Fig 1C and 1D**). We observed an increase in the percent of SG-positive cells as we increased the MOI during RSV cbVG-high infection, which correlates with the increased number of cbVG-containing particles in the inoculum. However, no differences in percent of SG-positive cells were observed between MOI 5 and MOI 10 (**Fig 1D**), suggesting that there is a threshold on the amount of SG-positive cells we can obtain at a given time during the infection. Taken together, these data indicate that presence of cbVGs during RSV infection correlates with SG formation.

### SGs form exclusively in cbVG-high cells during RSV cbVG-high infection

Using a previously described RNA fluorescence in situ hybridization (FISH)-based assay that allows differentiation of full-length genomes from cbVGs at a single-cell level [8], our lab

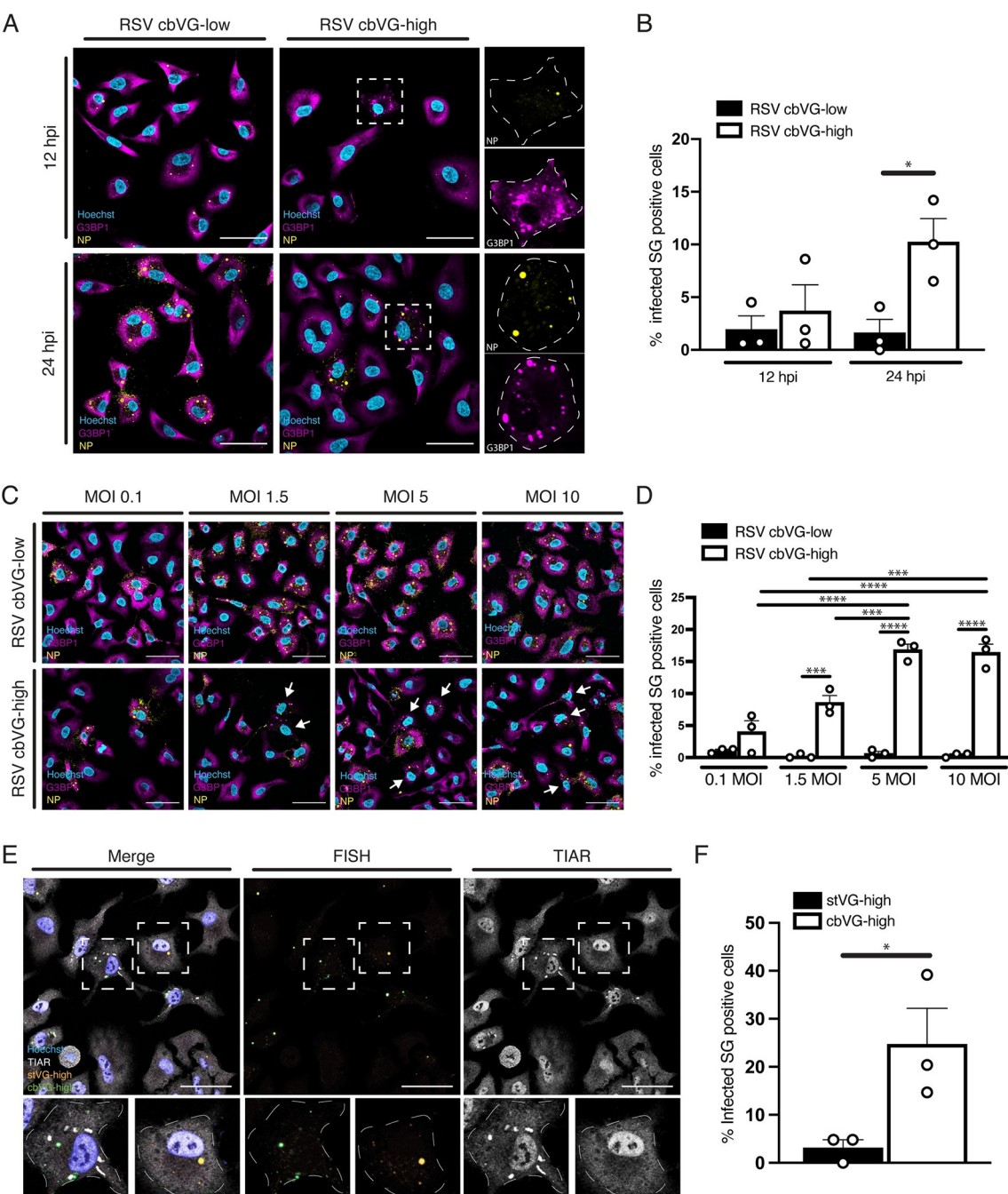

**Fig 1. SGs form during RSV infection containing high levels of cbVGs.** (**A**) SG (G3BP1, magenta) and viral protein (RSV NP, yellow) detection 12 and 24 hpi with RSV cbVG-high or cbVG-low virus at MOI of 1.5 $TCID_{50}$/cell. (**B**) Percent of SG-positive cells within the infected population 12 and 24 hpi with RSV cbVG-high and cbVG-low infections. Approximately 150 infected cells were counted per condition (average of 3 independent experiments shown). (**C**) SG (G3BP1, magenta) and viral protein (RSV NP, yellow) detection 24 hpi with RSV cbVG-high or cbVG-low at MOIs 0.1, 1.5, 5, and 10 $TCID_{50}$/cell. (**D**) Percent of SG-positive cells within the infected population 24 hpi with RSV cbVG-high and cbVG-low virus at MOIs 0.1, 1.5, 5, and 10 $TCID_{50}$/cell. Approximately 150 infected cells were counted per condition (average of 3 independent experiments shown). (**E**) SG detection (TIAR, white) in cells staining via FISH for stVG-high (orange) and cbVG (green) cells 24 hpi with RSV cbVG-high virus at MOI 1.5 $TCID_{50}$/cell. (**F**) Percent of SG-positive cells within the stVG-high and cbVG-high cell populations during RSV cbVG-high infection (average of 3 independent experiments shown). All widefield images were acquired with the Apotome 2.0 at 63× magnification and are representative of 3 independent experiments. Scale bar = 50 μm. Statistical analysis: one-way ANOVA (*$p < 0.05$, **$p < 0.01$, ***$p < 0.001$, ****$p < 0.00001$). Numerical data plotted can be found in the Supporting information: S1 Data. cbVG, copy-back viral genome; FISH, fluorescence in situ hybridization; G3BP1, GTPase-activating protein-binding protein 1; hpi, hours postinfection; MOI, multiplicity of infection; NP, nucleoprotein; RSV, respiratory syncytial virus; SG, stress granule; TIAR, TIA-1-related.

reported that cells infected with RSV or SeV cbVG-high stocks have heterogenous accumulation of viral genomes; some cells accumulate high levels of standard genomes (stVG-high), and others accumulate high levels of cbVGs (cbVG-high) [8,20,21]. To determine if SGs formed differentially within these 2 populations of cells, we combined RNA FISH with immunofluorescence to detect SGs during RSV cbVG-high infection. At 24 hpi, SGs formed almost exclusively in cbVG-high cells (green) and not stVG-high cells (orange) (**Fig 1E**). Interestingly, only around 30% of the cbVG-high cells had SGs (**Fig 1F**). This could suggest that a threshold of cbVG accumulation in the cells is needed for SG formation or that SG formation occurs asynchronously during infection, which is observed during HCV infection [22]. Nevertheless, these data demonstrate that cbVGs trigger SG formation.

## cbVGs induce SGs during SeV infection

To determine whether cbVG induction of SGs also occurs during infection with parainfluenza viruses, we infected cells with cbVG-high or cbVG-low SeV, a member of the paramyxovirus family and close relative to the human parainfluenza virus 1. Like infection with RSV, SGs formed predominantly during SeV cbVG-high infections (**Fig 2A**) where approximately 20% of the infected cells were positive for SGs at 24 hpi (**Fig 2B**). Compared to cells with undetected SGs or NP (**Fig 2A, right panel inset 1**), some SG-positive cells had notably low NP signal (**Fig 2A, right panel inset 2**), while other SG-positive cells showed high NP signal (**Fig 2A, right panel inset 3**).

To further establish the role of cbVGs in inducing SGs, we performed a dose-dependent experiment using purified cbVG-containing viral particles. We infected cells with SeV cbVG-low and supplemented the infection with increasing doses of purified cbVG-containing particles. The percent of SG-positive cells increased in proportion to the amount of purified cbVG-containing particles added (**Fig 2C, upper panel, and 2D**). SG were not observed, however, when we added the same amounts of UV-inactivated purified cbVG particles (**Fig 2C, lower panel, and 2D**). These data demonstrate that only replication-competent cbVGs induce SG formation during RNA virus infection.

## Canonical SGs are formed during cbVG-high infection

Some viruses can induce formation of SG-like granules that differ compositionally from canonical SGs and can relocalize SG components to viral replication centers [23–25]. Other viruses induce formation of RNAseL-dependent bodies (RLBs), which contain common proteins also found in SGs but are structurally and functionally distinct from SGs [26]. To better characterize the granules observed during RSV cbVG-high infection, we began by testing if cbVG-dependent granules require polysome disassembly, a crucial step for proteins to bind ribosome-free mRNA and form canonical SGs. For this, we treated RSV cbVG-high infected cells with cycloheximide (CHX), which inhibits canonical SGs by preventing polysome disassembly [27]. Sodium arsenite, a chemical known to induce canonical SGs, was used as a positive control [28]. Treatment with CHX during RSV cbVG-high infection led to a decrease in SG-positive cells compared to treatment with the drug's vehicle alone (DMSO) (**Fig 3A and 3B**). To rule out any effect the drugs could have on G3BP1 localization, we costained with another SG marker, TIA-1-related (TIAR) protein. Costaining with TIAR showed colocalization with G3BP1 in SGs in the DMSO-treated cells and disassembly from granules in the drug-treated conditions (**Fig 3A**), demonstrating that cbVG-dependent SGs are canonical SGs.

We next tested whether RSV-induced granules were RLBs [29]. To do this, we infected RNAseL knockout (KO) cells with RSV cbVG-high virus and looked at differences in SG formation comparing to poly I:C transfection, which is known to induce RLB formation [29].

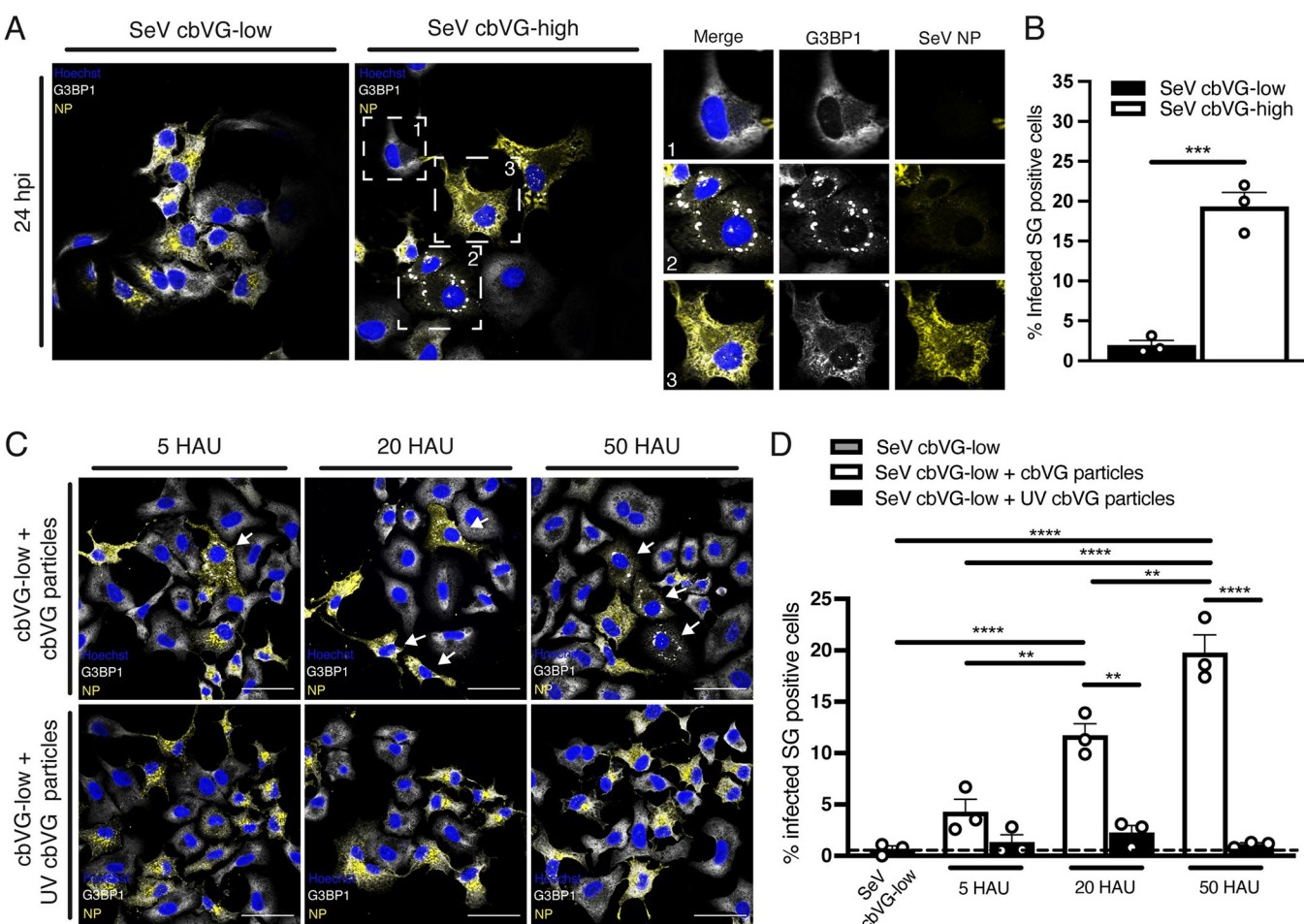

**Fig 2. SeV cbVGs induce SG formation.** (**A**) SG (G3BP1, white) and viral protein (SeV NP) detection 24 hpi with SeV cbVG-low and cbVG-high (NP, yellow) at MOI 1.5 $TCID_{50}$/cell. Digital zoomed images for each of the marked cells are shown in the panel on the right. (**B**) Percent of infected SG-positive cells 24 hpi with SeV cbVG-low and cbVG-high at MOI 1.5 $TCID_{50}$/cell. Approximately 150 infected cells were counted per condition (average of 3 independent experiments shown). (**C**) SG (G3BP1, white) and viral protein (SeV NP) detection 24 hpi at MOI 1.5 $TCID_{50}$/cell supplemented with either purified cbVG particles or UV-inactivated cbVG particles at increasing HAUs. (**D**) Percent of SG-positive cells at increasing HAU doses of active/UV-inactive cbVG particles. Approximately 200 infected cells were counted per condition (average of 3 independent experiments shown). All widefield images were acquired with the Apotome 2.0 at 63× magnification and are representative of 3 independent experiments. Scale bar = 50 μm. Statistical analysis: one-way ANOVA (*$p < 0.05$, **$p < 0.01$, ***$p < 0.001$, ****$p < 0.00001$). Numerical data plotted can be found in the Supporting information: S1 Data. cbVG, copy-back viral genome; G3BP1, GTPase-activating protein-binding protein 1; HAU, hemagglutination unit; hpi, hours postinfection; MOI, multiplicity of infection; NP, nucleoprotein; SeV, Sendai virus; SG, stress granule.

Structurally, RLBs are smaller, more punctate, and contain less TIAR than canonical SGs (**Fig 3C, left panel**). RNAseL activation prevents canonical SGs from forming by degrading free mRNA necessary for SGs to form and only when knocking out RNAseL can canonical SGs form upon stimulation [29,30]. SGs are structurally bigger and less uniform than RLBs. SGs formed during RSV cbVG-high infection even in RNAseL KO cells, and the structure of these granules was unchanged between cell lines, demonstrating that RSV-dependent SGs are not RLBs (**Fig 3C**).

We then investigated if, out of the context of an infection, cbVG RNA would still induce formation of canonical SGs or would induce RLBs similar to poly I:C. We transfected in vitro transcribed RSV and SeV cbVG-derived oligonucleotides that maintain the key stimulatory domains of cbVGs (RSV 238 and SeV 268 [9]) into A549 cells and compared to poly I:C-

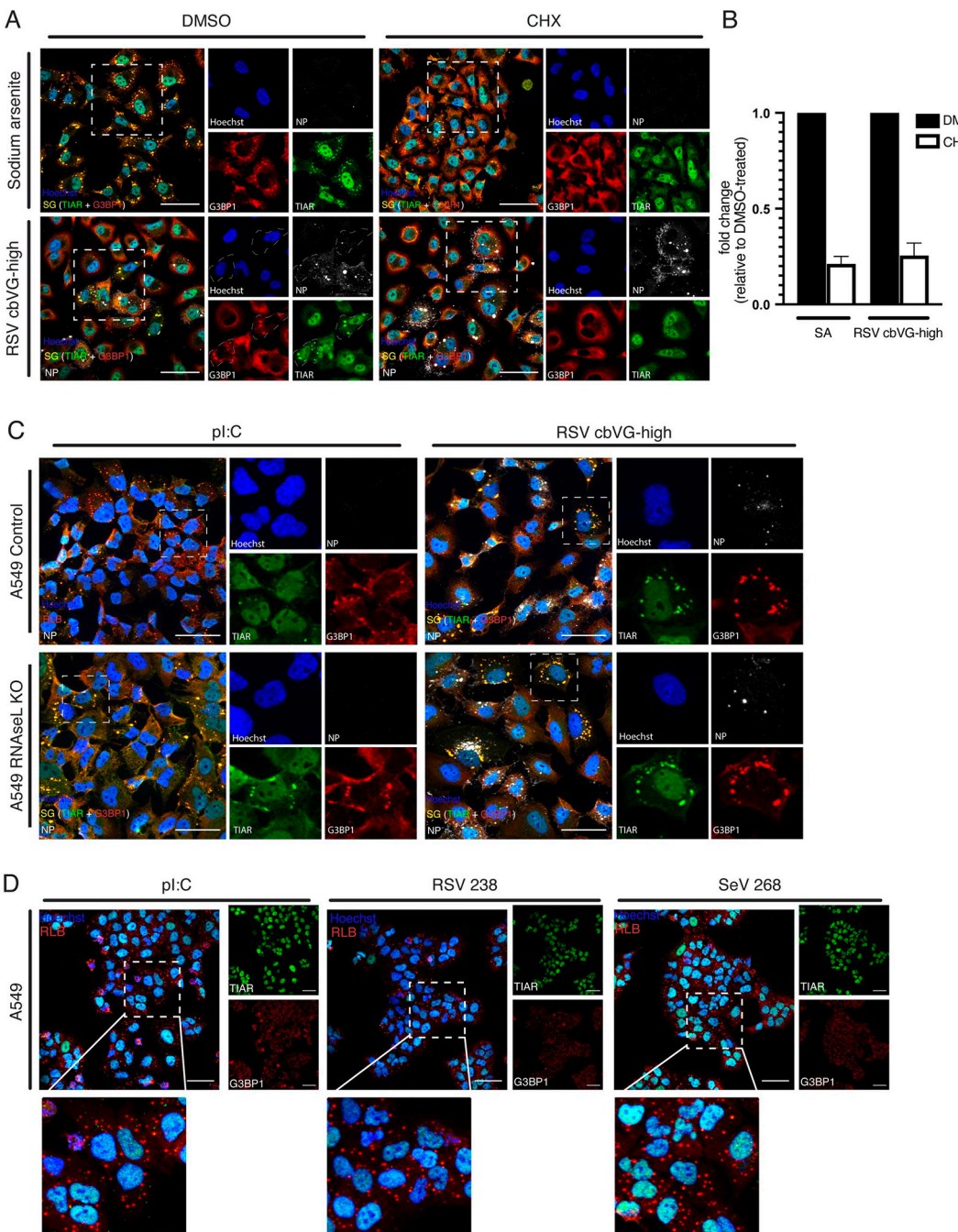

**Fig 3. RNA granules formed during RSV cbVG-high infection are canonical SGs.** (**A**) G3BP1 (red) and TIAR (green) staining for SGs in cells treated with SA (0.5 mM) for 1 h or infected with RSV cbVG-high (RSV NP, white) at MOI 1.5 $TCID_{50}$/cell 23 hpi and treated with DMSO or CHX (10 μg/mL) for 1 h. (**B**) Quantification of SG-positive cells after drug treatment in SA or RSV cbVG-high infected cells. Approximately 150 cells were counted for each condition. Fold change relative to DMSO-treated cells is shown from three independent experiments. (**C**) RNA granule detection (G3BP1, red; TIAR, green) in A549 control and RNAseL KO cells transfected with poly I:C 10 μg/mL or infected with RSV cbVG-high (RSV NP, white) 24 hpi at MOI 1.5 $TCID_{50}$/cell. (**D**) RNA granule detection (G3BP1, red; and TIAR, green) A549 cells transfected with poly I:C or RSV and SeV cbVG derived oligonucleotides RSV 238 and SeV 268. All widefield images were acquired with the Apotome 2.0 at 63× or 40× magnification. Scale bar = 50 μm. Numerical values plotted can be found in the Supporting information: S1 Data. cbVG, copy-back viral genome; CHX, cycloheximide; G3BP1, GTPase-activating protein-binding protein 1; hpi, hours postinfection; KO, knockout; MOI, multiplicity of infection; NP, nucleoprotein; RSV, respiratory syncytial virus; SA, sodium arsenite; SeV, Sendai virus; SG, stress granule; TIAR, TIA-1-related.

induced RLBs. We saw no differences in RNA granule formation and G3BP1 and TIAR contents between poly I:C RLBs and the granules observed with transfected cbVG-derived oligonucleotides (**Fig 3D**), indicating that cbVGs induce canonical SGs only in the context of SeV or RSV infection while RLBs are produced in response to naked cbVG RNA.

## cbVG-dependent SGs are PKR dependent and MAVS independent

To better understand the molecular mechanisms leading to SG formation in response to cbVGs during infection, we investigated the role of major dsRNA sensors in SG induction. SG formation during infection with many viruses, including RSV, depends on PKR activation [11]. To confirm that cbVGs induce PKR activation, we probed for PKR phosphorylation during RSV cbVG-high infection. As expected, PKR phosphorylation is increased during RSV cbVG-high infections compared to RSV cbVG-low or mock infection (**Fig 4A**). Because PKR is an IFN-stimulated gene (ISG) and cbVGs strongly induce the IFN response, higher levels of unphosphorylated PKR are expected during cbVG-high infection (**Fig 4A, middle blot**). To determine if cbVG-induced SGs are PKR dependent, we infected A549 PKR KO cells (**Fig 4B, middle lane**) and visualized SG formation. Consistent with the literature, PKR KO cells infected with RSV cbVG-high virus did not show SG-positive cells (**Fig 4D and 4E, middle**

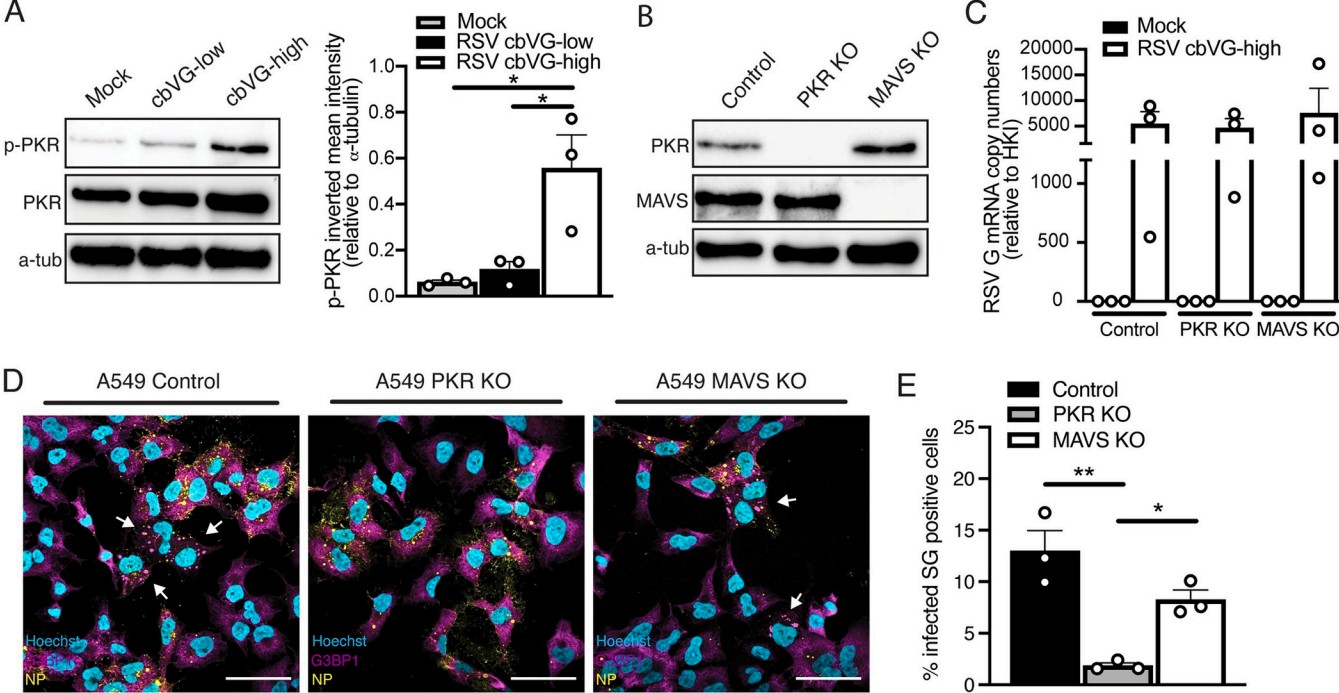

**Fig 4. cbVG-dependent SGs are PKR dependent and MAVS independent.** (**A**) Phosphorylation of PKR 24 hpi with RSV cbVG-low and cbVG-high at MOI 1.5 $TCID_{50}$/cell. p-PKR inverted mean intensity relative to α-tubulin is shown. Western blot images shown are representative of 3 independent experiments. Statistical analysis: one way ANOVA (*$p < 0.05$). (**B**) Western blot analysis showing efficient KO of PKR and MAVS in A549 cells. (**C**) Expression of *RSV G* gene mRNA relative to the HKI 24 hpi with RSV cbVG-high infection at MOI 1.5 TCID50/cell in control, MAVS, or PKR KO A549 cells (average of 3 independent experiments are shown). (**D**) SG (G3BP1, magenta) and viral protein (RSV NP) detection in PKR KO and MAVS KO A549 cells 24 hpi with RSV cbVG-high virus at MOI 1.5 $TCID_{50}$/cell. (**E**) Quantification of SG-positive cells 24 hpi with RSV cbVG-high at MOI 1.5 $TCID_{50}$/cell in PKR or MAVS KO A549 cells. Approximately 300 cells were counted per condition. All widefield images were acquired with the Apotome 2.0 at 63× magnification and are representative of 3 independent experiments. Scale bar = 50 μm. Statistical analysis: one-way ANOVA (*$p < 0.05$, **$p < 0.01$). Numerical values plotted can be found in the Supporting information: S1 Data. cbVG, copy-back viral genome; G3BP1, GTPase-activating protein-binding protein 1; HKI, housekeeping index; hpi, hours postinfection; KO, knockout; MAVS, mitochondrial antiviral signaling; MOI, multiplicity of infection; NP, nucleoprotein; RSV, respiratory syncytial virus; SG, stress granule.

**panel and bar**). *RSV G* mRNA levels were similar between cell types, confirming that inhibition of SGs in PKR KO cells was not due to lower replication of the virus (**Fig 4C, middle bar**). Together, these data suggest that the SGs observed during RSV cbVG-high infection are PKR dependent and that cbVG induction of SGs is mediated through PKR activation.

Because cbVGs exert most of their functions through RLR stimulation, which leads to MAVS activation and enhanced production of IFN, we sought to investigate whether cbVGs also induced SGs through MAVS signaling. To our surprise, MAVS KO cells (**Fig 4B, right lane**) infected with RSV cbVG-high virus showed SG-positive cells (**Fig 4D and 4E, right panel and bar**). The percent of SG-positive cells trended slightly lower than control but was not statistically significant (**Fig 4E**). This is most likely due to a reduced expression of PKR, a known ISG. Contrary to reports in the literature, we did not observe localization of MAVS in SGs (**S2A Fig**), nor recruitment of RIG-I to SGs during SeV cbVG-high infection (**S2B Fig**). These data indicate that cbVGs induce SGs independent of cbVGs immunostimulatory activity. To our knowledge, this is the first time cbVGs have shown to modulate cellular processes that are independent of MAVS signaling.

## cbVG-dependent SG inhibition requires KO of both G3BP1 and G3BP2

To form SGs, nucleating factors initiate RNA protein aggregation and liquid phase separation [10]. Studies suggest that one of these nucleating factors, G3BP1, is necessary and sufficient for SGs to form during viral infections [31–33]. To determine if G3BP1 is sufficient for cbVG-dependent SGs, we infected G3BP1 KO cells (**Fig 5A, second lane**) with RSV cbVG-high virus and looked at SGs using TIAR staining as proxy for SG formation. *RSV G* mRNA levels confirmed that there were not significant differences in viral replication between cell types (**Fig 5B**). Unexpectedly, we observed TIAR-containing SGs in G3BP1 KO cells (**Fig 5C, upper panel**). To confirm that these were canonical SGs and not aggregation of TIAR as an artifact of knocking out G3BP1, we treated the cells with CHX. Indeed, TIAR-containing SGs in G3BP1 KO cells are sensitive to CHX, suggesting that these were canonical SGs (**Fig 5C, lower panel**). These data indicate that knocking out G3BP1 is not sufficient to inhibit RSV-dependent SGs, contradicting what has previously been suggested in the literature [31].

In the context of some nonvirus-induced stresses, knocking out both G3BP1 and G3BP2 have shown to be necessary for SG inhibition [34]. To test if cbVG-dependent SG inhibition requires KO of both G3BP1 and G3BP2, we next generated a G3BP2 KO cell line as well as a G3BP1/2 double KO (dKO) cell line (**Fig 5A**). When we infected G3BP1/2 dKO cells with RSV cbVG-high virus stocks, we no longer observed SGs upon staining for TIAR, but SGs were still formed in G3BP1 and G3BP2 single KO cells (**Fig 5D**). These data demonstrate that cbVG-dependent SG inhibition requires KO of both G3BP1 and G3BP2.

## cbVG-dependent SGs are not required to induce the antiviral response

As SG formation is often associated with induction of the intrinsic antiviral immunity [12,14,16,17], we then determined if SGs are necessary for the expression of antiviral genes in response to cbVGs. To do this, we infected A549 control, G3BP1 KO, G3BP2 KO, and G3BP1/2 dKO cells with RSV cbVG-high and looked for differences in expression of genes involved in antiviral immunity, including IFNs and ISGs, at 24 hpi by qPCR. Expression of *IL-29*, *ISG56*, and *IRF7* mRNAs was not impaired when comparing control and G3BP1/2 dKO cells, and statistically significant differences in *IL-29* expression were only observed between G3BP1 KO and dKO (**Fig 6A–6C**).

To assess the impact of SGs on the host antiviral response more broadly, we looked at the whole transcriptome in A549 control and KO cells at 24 hpi. Most ISGs were expressed at

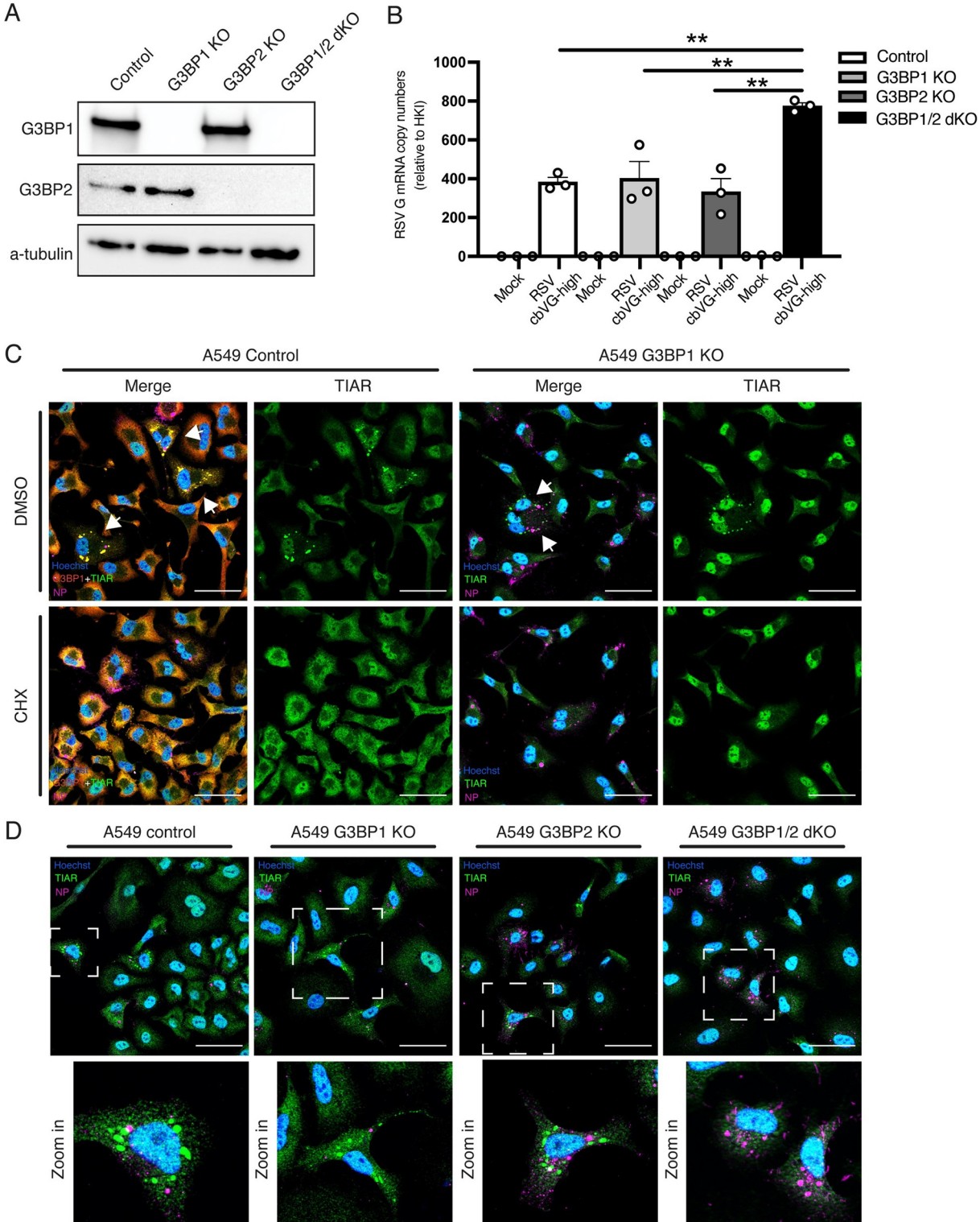

**Fig 5. cbVG-dependent SG inhibition requires KO of both G3BP1 and G3BP2.** (**A**) Western blot analysis validating A549 G3BP1 KO, G3BP2 KO, and G3BP1/2 dKO. (**B**) Expression of *RSV G* gene mRNA relative to the HKI 24 hpi with RSV cbVG-high infection at MOI 1.5 TCID$_{50}$/cell in A549 control, G3BP1 KO, G3BP2 KO, and G3BP1/2 dKO. (**C**) G3BP1 (red) and TIAR (green) staining for SG and viral protein (RSV NP) detection in control and G3BP1 KO cells 24 hpi with RSV cbVG-high at MOI 1.5 TCID$_{50}$/cell and treated with DMSO (upper panel) or CHX (10 μg/mL) (lower panel). (**D**) G3BP1 (red) and TIAR (green) staining for SGs and viral protein (RSV NP) detection in A549 control,

G3BP1 KO, G3BP2 KO, and G3BP1/2 dKO cells infected 24 hpi with RSV cbVG-high at MOI 1.5 TCID$_{50}$/cell. No SG-positive cells were detected in any field of G3BP1/2 dKO A549 cells. All widefield images were acquired with the Apotome 2.0 at 63× magnification and are representative of 3 independent experiments. Scale bar = 50 μm. Statistical analysis: one-way ANOVA (**$p < 0.01$). Numerical values plotted can be found in the Supporting information: S1 Data. cbVG, copy-back viral genome; CHX, cycloheximide; dKO, double KO; G3BP1, GTPase-activating protein-binding protein 1; HKI, housekeeping index; hpi, hours postinfection; KO, knockout; MOI, multiplicity of infection; NP, nucleoprotein; RSV, respiratory syncytial virus; SG, stress granule; TIAR, TIA-1-related.

similar levels in control and dKO cells (difference in expression were less than 2-fold; **Fig 6D**). In the few cases when there were differences of 2-fold decrease or more in expression, the difference was also observed in the G3BP1 or G3BP2 single KO conditions, suggesting that the difference is driven by processes independent of SG formation (**Fig 6D, right panel**). Additionally, we tested whether absence of SGs leads to reduced protein expression of ISGs. Expression of IFIT1, IRF7, and RIG-I was not different between the cell lines, demonstrating that the antiviral immune response is not dependent on SG formation (**Fig 6E and 6F**).

Because the role G3BPs have in the stress response is directly in SG formation and not the translation inhibition that occurs upstream of the pathway, we looked at the direct role of PKR signaling in antiviral immunity. For this, we infected PKR KO cells with RSV cbVG-high virus and compared *IL-29* transcript levels and IFIT1 protein levels to control infected cells and saw no significant differences (**Fig 6G and 6H**). Similarly, cells infected with SeV cbVG-high virus had no differences in phosphorylation of IRF-3, the primary transcription factor leading to type I IFN expression, nor differences in protein expression of the antiviral gene IFIT1 (**S3A and S3B Fig**). Altogether, these data suggest that PKR activation and SG formation are dispensable for global induction of antiviral immunity.

## SeV cbVG-dependent SGs form dynamically during infection and correlate with reduced viral protein expression

To study the dynamics of SG assembly and disassembly as well as assess the impact of SGs during infection, we generated G3BP1-GFP expressing A549 cells to visualize SG formation in real time. Using live-cell imaging of cells infected with a recombinant SeV expressing miRFP670 (rSeV-C$^{miRF670}$) and supplemented with purified cbVG particles, we show dynamic formation and disassembly of SGs throughout the course of the infection (**S1 Movie**). During the period of 6 to 72 hpi, we identified several subpopulations of cells (**Fig 7A**). Some cells formed SGs after infection and eventually disassembled them (**Fig 7A, series 1**). These cells showed faint levels of miRFP670 signal early in infection. Once SGs disassembled, the miRFP670 signal increased. Other cells formed SGs and eventually died (**Fig 7A, series 2**). A few cells assembled and disassembled SGs and remained very low in miRFP670 signal throughout the infection (**Fig 7A, series 3**). Moreover, formation of SGs persisted in the population even 13 dpi (**Fig 7B**). These data demonstrate that SeV cbVG-dependent SGs form asynchronously and that formation of SGs continues throughout the infection.

In these experiments, we observed that the signal for the viral reporter gene miRFP670 was low in SG-positive cells, to the point where some cells appeared uninfected. This is similar, but more extreme, than our observation via immunofluorescence that SeV NP-positive SG-positive cells often showed lower signal for SeV NP compared to those that were SG-negative cells (**Fig 2A**). We observed similar findings in RSV cbVG-high infection when staining for the RSV F protein (**S4 Fig**). We hypothesized that a single cell could gain and lose miRFP670 signal within a 6-h window, resulting in SG-positive cells that appeared uninfected at the time of imaging. To confirm that SG-positive cells during live imaging were infected, we performed time-lapse microscopy starting at 6 hpi before we begin to see SG-positive cells during the infection and tracked SG-positive cells every 30 min from 6 to 24 hpi to assess changes in the

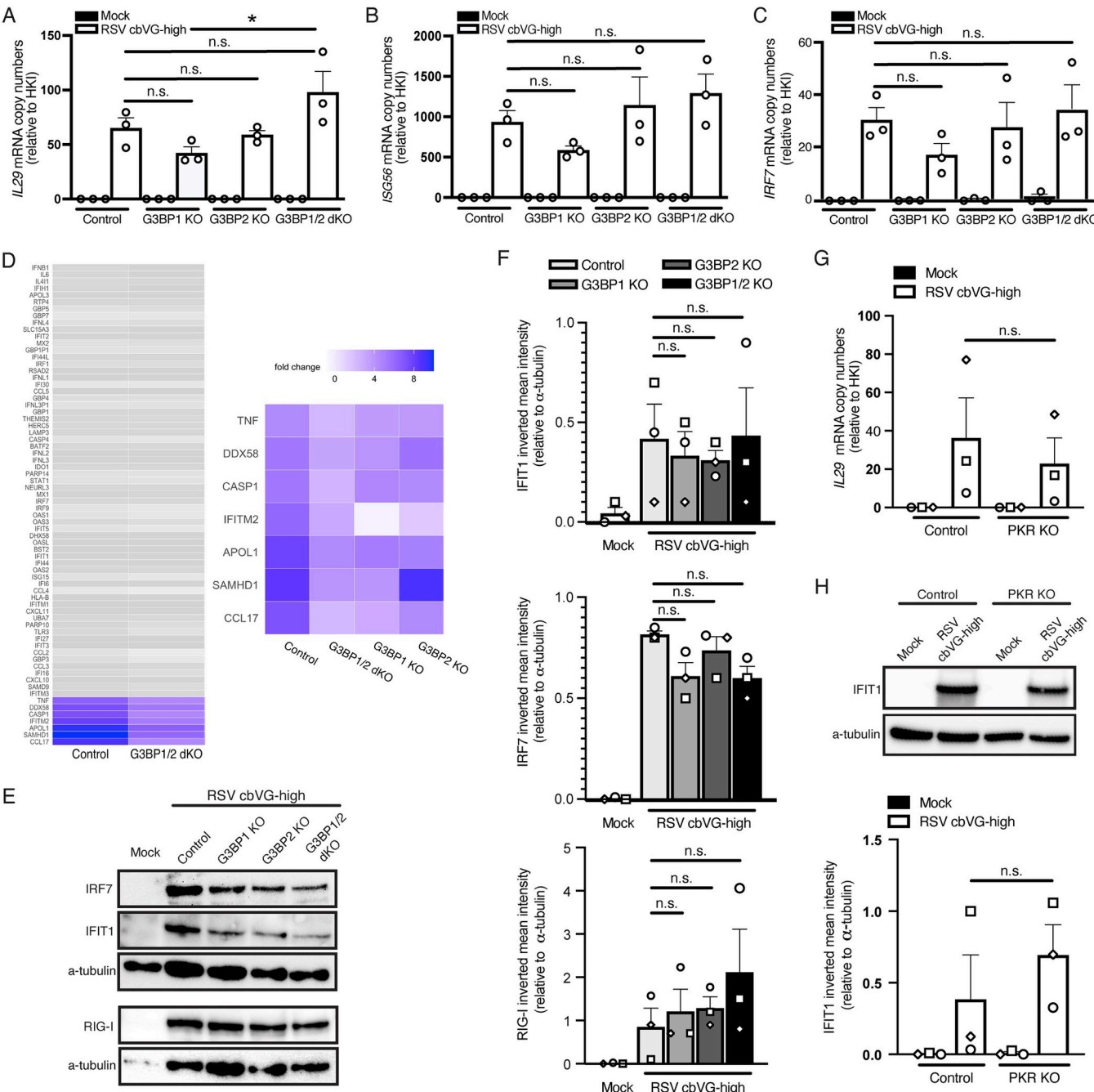

**Fig 6. cbVG-dependent SGs are not required for the antiviral response during RSV cbVG-high infection.** mRNA copy numbers of (**A**) *IL29*, (**B**) *ISG56*, and (**C**) *IRF7* relative to the HKI in A549 control, G3BP1 KO, G3BP2 KO, and G3BP1/2 dKO cells 24 hpi with RSV cbVG-high at MOI 1.5 TCID$_{50}$/cell. Statistical analysis: one-way ANOVA (*$p < 0.05$). (**D**) Log 2-fold change analysis of genes related to the antiviral response in A549 control, G3BP1 KO, G3BP2 KO, and G3BP1/2 dKO cells 24 hpi with RSV cbVG-high at MOI 1.5 TCID$_{50}$/cell relative to mock-infected cells. Genes that had less than a 2-fold decrease difference between control and G3BP1/2 dKO are represented in grey color. Genes that had more than a 2-fold decrease difference are highlighted in color. Genes with 2-fold decrease or more difference between control and G3BP1/2 dKO are shown in the right panel and compared to the log 2-fold change of G3BP1 and G3BP2 single KOs. (**E**) Western blot analysis of RIG-I, IFIT1, and IRF7 in A549 control, G3BP1 KO, G3BP2 KO, and G3BP1/2 dKO cells 24 hpi with RSV cbVG-high at MOI 1.5 TCID$_{50}$/cell. (**F**) Inverted mean intensity quantification of IRF7, IFIT1, and RIG-I western blot bands relative to α-tubulin loading control. Statistical analysis: one-way ANOVA. No statistical significance was found. (**G**) *IL29* mRNA levels relative to the HKI in control and PKR KO cells 24 hpi with RSV cbVG-high at MOI 1.5 TCID$_{50}$/cell. Statistical analysis: one-way ANOVA. No statistical significance was found. (**H**) Western blot analysis of IFIT1 and IRF7 in control and PKR KO cells 24 hpi with RSV cbVG-high at MOI 1.5 TCID$_{50}$/cell. Statistical analysis: one-way ANOVA. No statistical significance was found. All western blot images shown are representative of 3 independent experiments. Numerical values plotted can be found in the Supporting information: S1 Data. cbVG, copy-back viral genome; dKO, double KO; G3BP1, GTPase-activating protein-binding protein 1; HKI, housekeeping index; hpi, hours postinfection; KO, knockout; MOI, multiplicity of infection; RSV, respiratory syncytial virus; SG, stress granule.

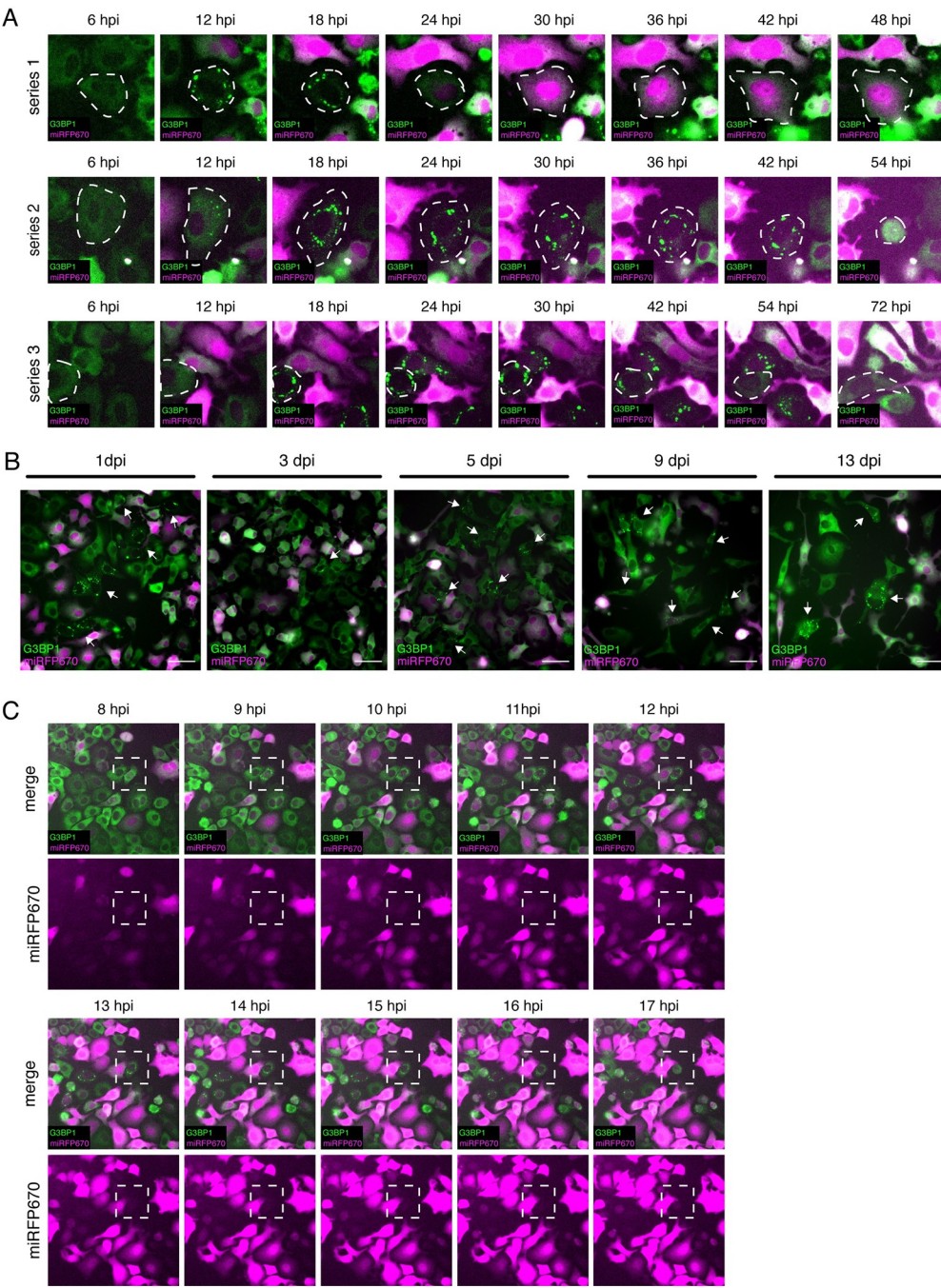

**Fig 7. SeV cbVG-dependent SGs form asynchronously and are maintained at the population level throughout the infection.** (**A**) G3BP1-GFP (green) expressing A549 cells infected with rSeV-C$^{miRF670}$ (magenta) reporter virus at MOI 3 TCID$_{50}$/cell with 20 HAU of supplemented cbVG purified particles, time-lapse microscopy 6–72 hpi, images every 6 h at a 20× magnification. Series show focus of different cells in the population. (**B**) Time-lapse microscopy images of G3BP1-GFP (green) expressing A549 cells infected with rSeV-C$^{miRF670}$ (magenta) reporter virus at MOI 3 TCID$_{50}$/cell with 20 HAU of supplemented cbVG purified particles from day 1 to day 13. (**C**) G3BP1-GFP (green) expressing A549 cells infected with rSeV-C$^{miRF670}$ (magenta) reporter virus at MOI 3 TCID$_{50}$/cell with 20 HAU of supplemented cbVG purified particles, time-lapse microscopy 8–18 hpi, images taken every 1 h. All time-lapse images were acquired with a widefield microscope at 20× magnification. cbVG, copy-back viral genome; G3BP1, GTPase-activating protein-binding protein 1; HAU, hemagglutination unit; hpi, hours postinfection; MOI, multiplicity of infection; SeV, Sendai virus; SG, stress granule.

miRFP670 signal with a higher temporal resolution. SG-positive cells showed miRFP670 before forming SGs and lost the signal as time went by, demonstrating that SG formation is correlated with a reduction in viral protein expression (**Fig 7C and S2 Movie**).

## cbVG-mediated interference with viral protein expression is independent of MAVS signaling

The reduction on viral protein levels in SG-positive cells led us to hypothesize that the well-established viral interference function of cbVGs is at least in part mediated by the induction of the cellular stress response. Because cbVGs are known to interfere with virus replication through the induction of MAVS signaling and IFN production, which consequently leads to a reduction of viral protein levels, we determined if this viral protein reduction observed in SG-positive cells was due to the IFN response and independent on SG formation. To test this, we infected MAVS KO cells with SeV cbVG-high and compared viral protein SeV NP expression to control infected cells. SG-positive cells showed similar SeV NP fluorescence in control and MAVS KO cells (**Fig 8A**). These data suggest that the interference in viral protein expression observed in cbVG and SG-positive cells is not due to the IFN response and, instead, suggest a direct role for the cellular stress response in reducing viral protein expression.

## cbVGs induce translation arrest in SG-positive cells, leading to reduced viral protein expression

SGs form because of translation inhibition, which could affect viral protein levels in SG-positive cells. To determine if translation is inhibited specifically in cbVG-induced SG-positive cells, we performed a ribopuromycylation assay to detect active translation at a single-cell level using puromycin (PMY) immunostaining. PMY mimics the tyrosine-modified tRNA and, upon exposure to cells, is added to nascent peptides. By combining with a translation elongation inhibitor to trap peptides into ribosomes and upon fixing and immunostaining with a PMY-specific antibody, we can detect translation at a single-cell level. We performed ribopuromycylation in SeV cbVG-high infected cells and compared PMY staining in SG-positive cells to SG-negative cells. A reduction of PMY signal was observed almost exclusively in SG-positive cells during SeV cbVG-high infection (**Fig 8B, lower panel, and Fig 8C**). This reduction in signal was comparable to sodium arsenite–treated cells (**Fig 8B, middle panel, and Fig 8C**). To determine if SG formation is necessary for translation inhibition and reduced viral protein expression, we performed ribopuromycylation in G3BP1/2 dKO and compared PMY staining with control-infected cells. As expected, we observed low PMY in G3BP1/2 dKO single cells (**S5 Fig**), demonstrating that SGs form as a consequence of translation inhibition and are not the drivers of translational arrest. Overall, these data highlight a new function of cbVGs in triggering translation inhibition and SG formation independent of their role in inducing the antiviral immune response.

## Discussion

cbVGs shape the outcome of SeV and RSV infections [3,5–8,35]. Their importance is highlighted by their involvement in inducing antiviral immunity, interfering with virus replication, and establishing persistent viral infections [6–8]. Here, we demonstrate yet another role for cbVGs: to induce translation inhibition by activating PKR signaling and SG formation. PKR activation by cbVGs is independent of the MAVS pathway, highlighting the ability of cbVGs to induce cellular pathways independent of their immunostimulatory activity. Because the content of cbVGs in viral stocks used in experiments is usually not characterized or

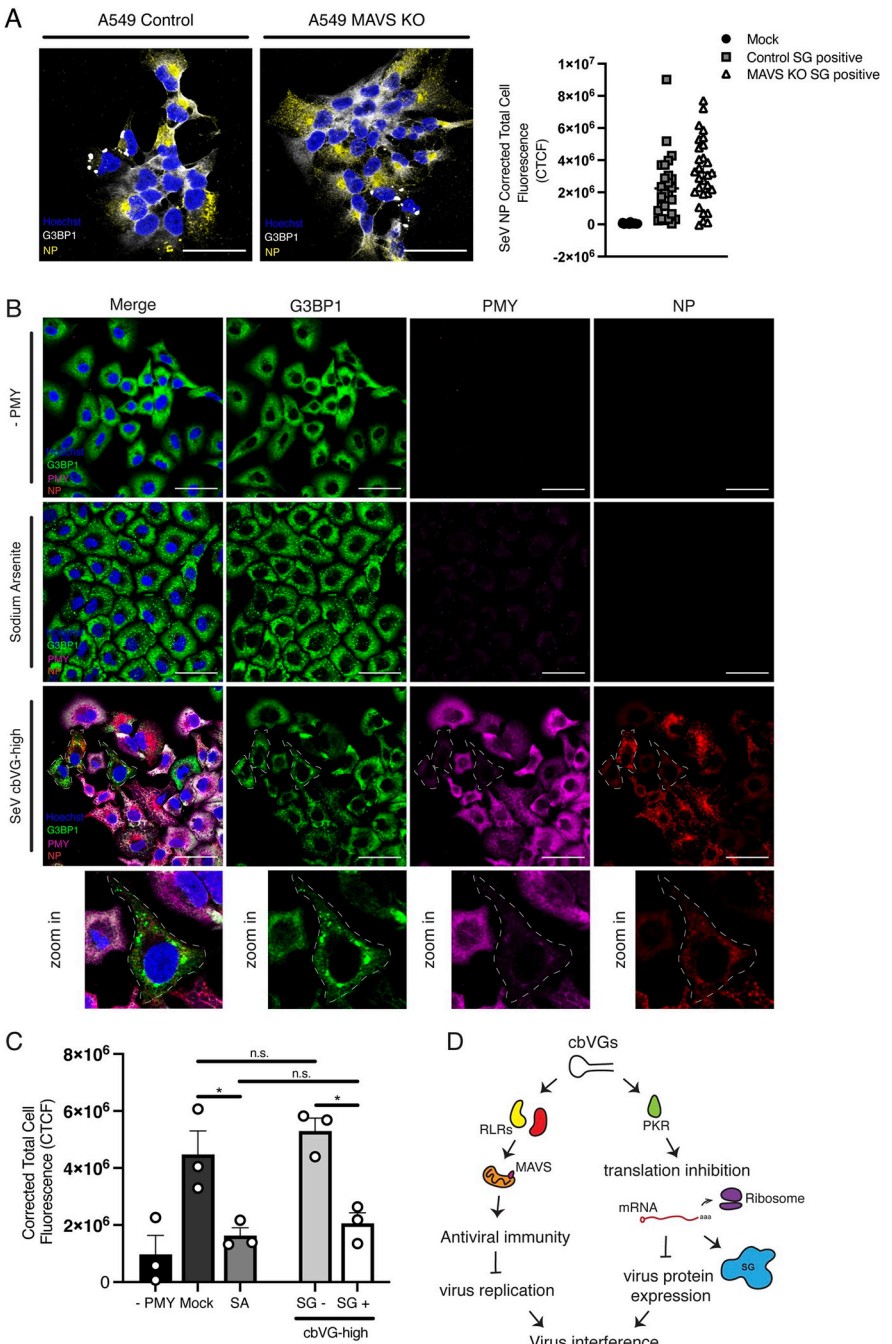

**Fig 8. cbVGs induce translation inhibition in SG-positive cells.** (**A**) SG (G3BP1 white) and viral protein (SeV NP) detection in control and MAVS KO A549 cells 24 hpi with SeV cbVG-high virus at MOI 1.5 TCID$_{50}$/cell 24 hpi. CTCF quantification of SeV viral protein NP in control and MAVS KO A549 SG-positive cells. (**B**) G3BP1 (green) for SG detection and PMY (magenta) for translation in cells infected with SeV cbVG-high (SeV NP, red) at MOI 3 TCID$_{50}$/cell 24 hpi or treated with sodium arsenite, with and without treatment with PMY for 5 min. (**C**) Quantification of PMY intensity (CTCF) in cells after drug treatment with SA or SG-positive and SG-negative SeV cbVG-high infected cells. Each dot represents the CTCF average of approximately 100 cells counted for each condition. Averages of three independent experiments are shown. Widefield images were acquired with the Apotome 2.0 at 63× magnification, scale bar = 50 μm. Statistical analysis: one-way ANOVA (*$p < 0.05$). (**D**) Diagram summarizing the role cbVGs have in inducing virus interference through activation of MAVS signaling and induction of translation inhibition. Numerical values plotted can be found in the Supporting information: S1 Data. cbVG, copy-back viral genome; CTCF, corrected total cell fluorescence; G3BP1, GTPase-activating protein-binding protein 1; hpi, hours postinfection; KO, knockout; MAVS, mitochondrial antiviral signaling; MOI, multiplicity of infection; NP, nucleoprotein; PMY, puromycin; SeV, Sendai virus; SA, sodium arsenite; SG, stress granule.

reported in the literature, our findings that cbVGs play a critical role in the induction of SGs help clarify contradicting evidence in the literature related to SG formation and function.

During virus infection, SGs can be inhibited by knocking out PKR, inhibiting both translation inhibition and SG formation, or by knocking out nucleating factors like G3BP1, which only inhibits the physical formation of SGs and not translation inhibition [11,36–38]. To test the impact of SG formation during cbVG-high infection, we generated a G3BP1 KO model that was previously shown to be required for SG formation during RSV infection [31]. To our surprise, knocking out G3BP1 was not sufficient to inhibit the formation of canonical SG containing TIAR (**Fig 5C**). After demonstrating that knocking out G3BP2 was also insufficient to inhibit SG formation, we developed a G3BP1 and G3BP2 dKO cell line, which successfully stopped SG from forming during RSV cbVG-high infection (**Fig 5D**). Our data agree with a recent report showing that knocking out both G3BP1 and G3BP2 is required for SG inhibition during viral infections [39]. We attribute the contradiction regarding the requirement of G3BP1 for SG formation during infection with mononegavirales to differences in the approaches used to validate the absence of SG, and staining for TIAR represents a good alternative SG marker for this purpose.

The immunostimulatory ability of cbVGs together with published data demonstrating how SG are involved in inducing and sustaining the immune antiviral response [13,14,32,40] led us to hypothesize that cbVGs induced SG to aid with activation of the antiviral response. However, using a G3BP1/2 dKO cell line that successfully impeded the formation of SG, we observed no differences in the induction of the antiviral response relative to the control cell line (**Fig 6**). This applied to both the transcriptional and the translational level of IFN and ISG expression (**Fig 6A–6C, 6E and 6F**). A broader transcriptome analysis confirmed that inhibition of SG during infection does not hamper the antiviral immune response (**Fig 6D**). Certain genes were down-regulated in the single G3BP1 or G3BP2 KO, which indicate SG-independent roles for these proteins during infection. Indeed, G3BP1 is reported to have roles involving the antiviral response [41]. We also determined if the effect the cellular stress response had on the global antiviral response was dependent on the translation inhibition that occurred upstream of SG formation. For this, we used the PKR KO system to look at how expression of antiviral genes and proteins were affected. Like the G3BP1/2 dKO system, we observed no differences in IFN and ISG expression in both RSV and SeV infection (**Figs 6G, 6H and S4**), suggesting that the cellular stress response does not impact the overall antiviral response during RSV or SeV infection.

By performing live-cell imaging of G3BP1-GFP expressing cells infected with a reporter SeV, we uncovered interesting facts about the dynamic formation and disassembly of SG. First, we observed that cbVG-induced SG are dynamic and form asynchronously throughout the infection (**S1 Movie**). Although we began observing SG-positive cells after 8 hpi, the number of SG-positive cells increased over time (**S1 Movie**). We could also see some cells disassembling SG throughout the infection (**Fig 7A, series 1 and 3**). These data likely explain why not all cbVG-high cells are SG positive when we look at a singular time point 24 hpi (**Fig 1E and 1F**). Although we do not have a working tool in our lab to perform RNA FISH in live cells, we suspect that all cbVG-high cells form SG at some point during the infection, but these are not all captured in snapshot immunofluorescence experiments. The mechanisms governing timing and maintenance of SG formation in a cbVG-positive cell remains unknown. We speculate that there could be a threshold level of cbVG amounts inside the cell that is required to trigger SG formation. Perhaps cell state factors could also be involved in controlling SG formation, such as cell cycle state, circadian rhythm phase, or even metabolic state.

Throughout our studies, we observed a drastic reduction in the viral protein level in SG-positive cells (**Figs 2A and S3 and S1 Movie**). We reasoned that the reduction in viral protein

expression could be explained by the well-known function of cbVGs in interfering with the virus life cycle via MAVS and expression of antiviral proteins. In contrast, we saw a reduction of viral protein levels even MAVS KO SG-positive cells (**Fig 8A**), suggesting that the interference in protein expression observed in SG-positive cbVG-high cells is regulated by activation of the stress response itself. Indeed, we observe drastic translation reduction at a single-cell level in SG-positive cells infected with cbVG-high virus (**Fig 8B and 8C**). These data implicate translation inhibition and SG formation as a previously undescribed mechanism of cbVG-mediated viral interference. We do not see differences in *RSV G* mRNA in control and PKR KO cells (**Fig 4C**), suggesting that PKR activation and translation inhibition does not interfere with virus transcription and, instead, interferes with the virus at the translational level due to the translation arrest accompanied by SG formation. We cannot, however, discard a potential additional role for cbVGs in directly interfering with the virus by competing with the virus polymerase, thereby reducing replication, transcription, and translation. However, our group has previously shown that this mechanism of interference is minimal [6]. Finally, detection of SGs in cells several days after infection extends the role of translation inhibition and SG formation to later phases of the infection (**Fig 7B**). It remains unknown if and which other proteins are affected by cbVG-dependent translation inhibition. Although we cannot rule out that a reduction on ISG protein expression occurs specifically in SG-positive cells, this potential reduction does not affect the global immune antiviral response during the infection, contradicting reports of a requirement for SGs for the initiation of antiviral immunity [14,15,17,39].

The detailed effects of viral protein reduction in RSV and SeV cbVG-high infections remain to be defined, as does the detailed mechanism of how cbVGs induce translation inhibition and SG formation. As SG-positive cells are only 10% to 20% of the infected population at a given time, developing tools to isolate and sort SG-positive cells from the rest of the population is essential to further understand the role translation inhibition and SG formation play.

Our work highlights how cbVGs, a subset of the virus genome population, are responsible for shaping the cellular stress response during negative-sense RNA virus infection. Although it was previously thought that SGs played a role in inducing global antiviral immunity, our data instead suggest a role for SGs in interfering with viral protein translation because of translation inhibition in SeV and RSV infections with high levels of cbVGs. This role extends to later phases of the infection and suggests translation inhibition and SG formation are important both in acute and persistent infections.

## Materials and methods

### Cell lines and viruses

A549 (human type II pneumocytes; ATCC CCL-185) cells were cultured in tissue culture medium (Dulbecco's Modified Eagle's Medium [DMEM; Invitrogen]) supplemented with 10% fetal bovine serum (FBS), gentamicin 50 ng/ml (Thermo Fisher), L-glutamine 2 mM (Invitrogen), and sodium pyruvate 1 mM (Invitrogen) at 5% $CO_2$ 37°C. The generation of A549 CRISPR KO cell lines has been described previously [42,43]. Plasmids used for CRISPR KO cells lines lentiCRISPR v2 and lentiCRISPR v2-Blast were originally from Feng Zhang (Addgene plasmid # 52961; http://n2t.net/addgene:52961; RRID: Addgene_52961) and Mahan Babu (Addgene plasmid # 83480; http://n2t.net/addgene: 83480; RRID: Addgene_83480), respectively. All KO cell lines were single-cell cloned, and confirmed KO for each clone was tested through western blot analysis. For generation of A549 G3BP1-GFP, we used the phage UbiC G3BP1-GFP-GFP plasmid originally from Jeffrey Chao (Addgene plasmid # 119950; http://n2t.net/addgene: 119950; RRID: Addgene_119950) [44]. G3BP1-GFP expressing cell lines were single-cell cloned. Cells were treated with mycoplasma removal agent (MP

Biomedical) and tested monthly for mycoplasma contamination using MycoAlert Plus mycoplasma testing kit (Lonza). SeV Cantell strain was grown in 10-day-old, embryonated chicken eggs (Charles River) for 40 h as previously described [45]. RSV stocks were grown in Hep2 cells as previously described [19] and harvested by collecting the cells supernatant. SeV and RSV cbVG-high and cbVG-low stocks were produced and characterized as described previously [19].

## Virus infections

For RSV infections, cells were washed once with PBS and then incubated with virus suspended in tissue culture medium supplemented with 2% FBS at 37°C for 2 h. Cells were then supplemented with additional 2% FBS tissue culture medium. For SeV infections, cells were washed twice with PBS and then incubated with virus suspended in infectious medium (DMEM; Invitrogen) supplemented with 35% bovine serum albumin (BSA; Sigma-Aldrich), penicillin-streptomycin (Invitrogen), and 5% $NaHCO_3$ (Sigma-Aldrich) at 37°C for 1 h. Cells were then supplemented with additional infectious medium. SeV cbVG particles were purified from the allantoic fluid of SeV infected embryonated eggs by density ultracentrifugation on a 5% to 45% sucrose gradient, as described previously [7].

## In vitro transcription of cbVGs

The pSL1180 plasmid was cloned to encode the SeV or RSV cbVGs as previously described [9]. The cbVG plasmid was linearized and in vitro transcribed using the MEGAscript T7 kit (Thermo Fisher). The resulting product was DNAse treated and purified by LiCl precipitation according to the manufacturer's protocol. All IVT RNA was quantified by Qubit (Thermo Fisher) and quality checked through Bioanalyzer (Agilent) to ensure a single band of the correct RNA length was obtained. For transfections, 5 pmol of the IVT cbVG or low molecular weight polyinosine-polycytidylic acid (poly I:C, InvivoGen) were transfected into control A549 or RNAseL-KO A549 cells. At 6 hours posttransfection, cells were fixed, permeabilized, and immunostained as described below for immunofluorescence.

## Recombinant Sendai virus rSeV-C$^{miRF670}$ rescue

To create the pSL1180-rCantell plasmid, the complete viral genome of the SeV Cantell strain and the necessary regulatory elements were inserted into the pSL1180 vector using SpeI and EcoRI restriction enzymes in the following order: T7 promoter, Hh-Rbz, viral genome, Ribozyme, and T7 terminator. A miRF670 gene was then inserted between the NP and P genes in the pSL1180-rCantell plasmid to create the pSL1180-rCantell-miRF670 plasmid. The noncoding region between the NP and P genes was used to separate the NP, miRF670, and P genes. Additional nucleotides were inserted downstream of the miRF670 gene to ensure that the entire genome followed the "rule of six." The NP, P, and L genes of Cantell were cloned into the pTM1 vector to generate the 3 helper plasmids. All plasmids were validated by sequencing. The recombinant virus rSeV-C$^{miRF670}$ was produced by cotransfecting pSL1180-rCantell-miRF670 and the 3 helper plasmids. BSR-T7 cells were transfected with a mixture of plasmids containing 4.0 μg pSL1180-rCantell-miRF670, 1.44 μg pTM-NP, 0.77 μg pTM-P, and 0.07 μg pTM-L using Lipofectamine LTX. After 5 h, the medium was replaced with infection medium containing 1 μg/ml TPCK, and the cells were incubated at 37°C. The expression of miRF670 was monitored daily using fluorescence microscopy. At 4 days posttransfection, the cell cultures were harvested, and the supernatants were used to infect 10-day-old specific pathogen-free embryonated chicken eggs via the allantoic cavity after repeated freeze-thaw cycles. After 40 h of incubation, the allantoic fluid was collected.

## Immunofluorescence

Cells were seeded at $1 \times 10^5$ cells/mL confluency in 1.5 glass coverslips (VWR) a day prior infection or drug treatment. The coverslips were transferred to a fresh plate and washed with PBS. Cells were fixed on the coverslips using 4% paraformaldehyde (EMS) for 15 min. Cells were then permeabilized with 0.2% Triton X-100 (Sigma-Aldrich) for 10 min. Primary and secondary antibodies diluted in 3% FBS were added and incubated for 1 h and 45 min, respectively. The nuclei were stained with a 1:10,000 dilution of Hoechst 33342 (Invitrogen) in PBS for 5 min prior to mounting. Coverslips were mounted in slides using Prolong Diamond antifade mounting media (Thermo Fisher) and curated overnight at room temperature. Antibodies used: SeV NP (clone M73/2, a gift from Alan Portner, directly conjugated with DyLight 594 or 647 *N*-hydroxysuccinimide (NHS) ester (Thermo Fisher)), RSV NP (Abcam catalog number ab94806), G3BP (Abcam catalog numbers ab181150 and ab56574), G3BP2 (Cell Signaling catalog number 31799), TIAR (Santa Cruz catalog number sc-398372), Puromycin (Merck, catalog number MABE343).

## RNA FISH combined with immunofluorescence

Cells were seeded at $1 \times 10^5$ cells/mL confluency in 1.5 glass coverslips a day prior infection. The coverslips were transferred to a fresh plate and washed with sterile PBS. Cells were fixed in the coverslips using 4% formaldehyde (Thermo Fisher) for 10 min and permeabilized with 70% ethanol for 1 h at room temperature. Cells were incubated with primary and secondary antibodies diluted in 1% BSA (Sigma-Aldrich) containing RNAse OUT (Thermo Fisher) for 45 and 40 min, respectively. Cells were postfixed with 4% formaldehyde and washed with 2× SSC (Thermo Fisher) followed by wash buffer (2× SSC and 10% formamide in water). Cells were hybridized with 2.5 nM RSV-specific LGC Biosearch custom probes (See **Table 1**) conjugated to Quasar 570 or Quasar 670. Slides were incubated overnight at 37°C in a humidified chamber for hybridization. Cells were washed twice with wash buffer for 30 min each and once with 2× SSC for 5 min. Coverslips were mounted using ProLong Diamond Antifade mounting media and curated overnight. Slides were imaged using Zeiss Axio observer widefield microscope.

## RNA extraction and PCR/qPCR

RNA was extracted using TriZol reagent (Life Technologies). For qPCR, mRNA was reverse transcribed using high-capacity RNA to cDNA kit (Thermo Fisher). qPCR was performed using SYBR green (Thermo Fisher) and 5 μM of reverse and forward primers for genes *IL-29* (CGCCTTGGAAGAGTCACTCA and GAAGCCTCAGGTCCCAATTC); *ISG56* (GGATTCTGTACA ATACACTAGAAACCA and CTTTTGGTTACTTTTCCCCTATCC); *IRF7* (GATCCAGTCCCAA CCAAGG and TCTACTGCCCACCCGTACA) and *RSV G* (AACATACCTGACCCAGAATC and GGTCTTGACTGTTGTAGATTGCA) on an Applied Biosystems QuantStudio 5 machine. Relative mRNA copy numbers were calculated using relative delta CT values and normalized using a housekeeping index with GAPDH and β-actin. For PCR detection of cbVGs, viral RNA was reverse transcribed using a SuperScript III first-strand synthesis (Invitrogen) with Primer GGTGAGGAATCTATACGTTATAC for SeV and primer CTTAGGTAAGGATATGTA-GATTCTACC for RSV. PCR was then performed with Platinum Taq DNA polymerase (Invitrogen) with the reverse transcription primers and primer ACCAGACAAGAGTTTAAGAG ATATGTATT for SeV and primer CCTCCAAGATTAAAATGATAACTTTAGG for RSV. Bands were analyzed using gel electrophoresis.

**Table 1. Probe sequences for RSV negative-sense RNA genome probes.**

| Probe sequence | Probe name |
| --- | --- |
| gtgctctatcatcacagatc | RSV genome_1 |
| ccctagaaattacatgccat | RSV genome_2 |
| ggactacgtttctatcgtga | RSV genome_3 |
| acttatccttctttgttgga | RSV genome_4 |
| ataagtggagctgcagagtt | RSV genome_5 |
| attgtgtcatgctatggcaa | RSV genome_6 |
| ttcttcccacaagctgaaac | RSV genome_7 |
| cagaggatggtactgtgaca | RSV genome_8 |
| ggcgtaactacacctgtaag | RSV genome_9 |
| tagtgctctgagaactggtt | RSV genome_10 |
| agcttcaacaacaccaggag | RSV genome_11 |
| tattcatagcctcggcaaac | RSV genome_12 |
| ttgagttaccaagagctcga | RSV genome_13 |
| ccacacaccatacagaatca | RSV genome_14 |
| tcttcacttcaccatcacaa | RSV genome_15 |
| ccacacaccatacagaatca | RSV genome_16 |
| tcttcacttcaccatcacaa | RSV genome_17 |
| ttggaagcacacagctacac | RSV genome_18 |
| taccatatgcgctaatgtgt | RSV genome_19 |
| aatcatctatgccagcagat | RSV genome_20 |
| ggacagatctggtcttacag | RSV genome_21 |
| caaccatggctcttagcaaa | RSV genome_22 |
| agacaggccacatttacatt | RSV genome_23 |
| attgctctcaacctaatggt | RSV genome_24 |
| tggctaaggcagtgatacat | RSV genome_25 |
| agagatgggcagcaattcat | RSV genome_26 |
| tctaattggtttatatgtgt | RSV DVG_1 |
| gttaaacagcttgacaacca | RSV DVG_2 |
| ctacatatccttacctaagt | RSV DVG_3 |
| aaccatttatatatggtaga | RSV DVG_4 |
| gaagttttcagcaataaact | RSV DVG_5 |
| gtgttgttagtggagatata | RSV DVG_6 |
| actgcattgtcaaaactaaa | RSV DVG_7 |
| ataaagagtctattgatgca | RSV DVG_8 |
| atgctaaattgatactatca | RSV DVG_9 |
| ttcccagtatttaatgtagt | RSV DVG_10 |
| attacaataggtcctgcgaa | RSV DVG_11 |
| gggatcggaggtttacttag | RSV DVG_12 |

## Drug treatments

For sodium arsenite treatment, cells were washed once with PBS and replaced with fresh media containing 0.5 mM of sodium arsenite (Sigma-Aldrich) for 1 h at 37˚C. For CHX treatment, infected cells or cells treated with sodium arsenite were treated with 10 μg/mL of CHX for 1 h at 37˚C.

## Ribopuromycylation

Detection of protein translation at a single-cell level was adapted from the previously described puromycylation method [46]. In brief, cells were seeded at $1 \times 10^5$ cells/mL confluency in 1.5 glass coverslips (VWR) a day prior infection or drug treatment. After 24 hpi or 30 min post drug treatment, the media was replaced with PMY labeling medium containing 91 μM of PMY (Sigma-Aldrich) and 45 μM of emetine (Sigma-Aldrich) in tissue culture media and incubated for 5 min at 37˚C. The cells were placed on ice and washed with 1 mL of ice-cold PBS. For PMY removal, the PBS was replaced with extraction buffer containing 0.015% digitonin (Thermo Fisher), 50 mM, Tris-HCl (pH 8), 5 mM $MgCl_2$, 25 mM KCl, Halt Protease Inhibitor Cocktail (Thermo Fisher), 10 U/mL RNase Out (Thermo Fisher) and incubated for 2 min on ice. The extraction buffer was carefully removed and replaced with ice-cold wash buffer containing 50 mM, Tris-HCl (pH 8), 5 mM $MgCl_2$, 25 mM KCl, Halt Protease Inhibitor Cocktail (Thermo Fisher), and 10 U/ mL RNase Out (Thermo Fisher). The cells were then fixed with 4% paraformaldehyde (EMS) for 15 min at room temperature. Finally, immunostaining for PMY together with G3BP1 and virus protein NP was performed following the immunofluorescence protocol described above.

## Imaging analysis

SG quantification was performed using Aggrecount automated image analysis as previously described [47]. CTCF was analyzed using Fiji software. In brief, cell boundaries were defined with adjusted thresholds using the G3BP1 signal. Then, the CTCF was calculated using the formula: Integrated Density − (Area of selected cell × Mean fluorescence of background).

## Western blots

Cells were seeded at $2 \times 10^5$ cells/mL confluency a day prior infection or drug treatment. Protein was extracted using 1% NP 40 (Thermo Fisher) with 2 mM EDTA, 150 mM NaCL (Thermo Fisher), 5 mM Tris-HCl, 10% glycerol (Sigma-Aldrich), Halt Protease Inhibitor Cocktail (Thermo Fisher), and PhosSTOP (Sigma-Aldrich). After samples were incubated on ice for 20 min and centrifuged for 20 min at 4˚C, supernatant was transferred to new tubes and protein concentration was quantified using the Pierce BCA Protein Assay Kit (Thermo Fisher). Protein (10 to 25 μg) was denatured for 5 min at 95˚C, loaded in a 4% to 12% Bis Tris gel (Bio-Rad), and transferred to a PVDF membrane (Millipore Sigma). Membranes were incubated overnight with primary antibodies diluted in 5% BSA in TBS (Fisher) with 0.1% Tween20 (Sigma-Aldrich). Membranes were incubated with secondary antibodies (anti-mouse or anti-rabbit) conjugated with HRP for 1 h in 5% BSA in TBST. Membranes were developed using Lumi-light western blot substrate (Roche) to detect HRP and a ChemiDoc (Bio-Rad). Antibodies used for western blot: PKR (Cell Signaling catalog number 12297), p-PKR (Abcam catalog number ab32036), MAVS (Cell Signaling catalog number 3993), G3BP (Abcam catalog numbers ab181150 and ab56574), G3BP2 (Cell Signaling catalog number CS 31799), IFIT1 (Cell Signaling catalog number CS 12082S), IRF7 (Cell Signaling catalog number CS 4920S), RIG-I (Santa Cruz catalog number sc-98911), α-tubulin (Abcam catalog number ab52866).

## Statistics

Statistics were calculated using GraphPad Prism Version 9.

## Supporting information

**S1 Fig. Characterization of RSV cbVG-high and cbVG-low virus stocks.** (**A**) Diagram showing how cbVGs form during negative-sense single-stranded RNA virus infection. (**B**) Agarose

gel of cbVG PCR amplicons from A549 cells 24 hpi with RSV cbVG-high virus at MOI 1.5 $TCID_{50}$/cell. (**C**) Expression of *RSV G* and *IL-29* mRNAs in A549 cells 24 hpi with RSV cbVG-high or cbVG-low virus at MOI 1.5 $TCID_{50}$/cell. Statistical analysis: one-way ANOVA (*$p < 0.05$, **$p < 0.01$). Numerical values plotted can be found in the Supporting information: S1 Data. cbVG, copy-back viral genome; hpi, hours postinfection; MOI, multiplicity of infection; RSV, respiratory syncytial virus.
(TIF)

**S2 Fig. SG-positive cells do not show MAVS or RIG-I localization in SG during SeV cbVG-high infection.** (**A**) SG (G3BP1, magenta) and MAVS (yellow) staining in A549 cells 24 hpi with RSV cbVG-high virus MOI 1.5 $TCID_{50}$/cell. Zoomed in images of SG-positive cells are shown on the right with merge and MAVS and G3BP1 single channel. (**B**) SG (G3BP1, magenta) and RIG-I (yellow) staining in A549 cells 24 hpi with RSV cbVG-high virus MOI 1.5 $TCID_{50}$/cell. Zoomed in images of SG-positive cells are shown on the right with merge and RIG-I and G3BP1 single channel. Widefield images at 40× magnification. cbVG, copy-back viral genome; G3BP1, GTPase-activating protein-binding protein 1; hpi, hours postinfection; MAVS, mitochondrial antiviral signaling; MOI, multiplicity of infection; RIG-I, retinoic acid–inducible gene I; RSV, respiratory syncytial virus; SeV, Sendai virus; SG, stress granule.
(TIF)

**S3 Fig. The stress response is dispensable for overall antiviral immunity during SeV cbVG-high infections.** (**A**) Western blot analysis of phosphorylated IRF-3 in control and PKR KO cells 6 and 18 hpi with SeV cbVG-high at MOI 1.5 $TCID_{50}$/cell. (**B**) Western blot analysis of IFIT1 in A549 control and PKR KO cells 24 hpi with SeV cbVG-high at MOI 1.5 $TCID_{50}$/cell. IFIT1 inverted mean intensity relative to α-tubulin is shown. Images shown are representative of 2 independent experiments. Numerical values plotted can be found in the Supporting information: S1 Data. cbVG, copy-back viral genome; hpi, hours postinfection; KO, knockout; MOI, multiplicity of infection; SeV, Sendai virus.
(TIF)

**S4 Fig. SG-positive cells show reduced RSV F protein expression during RSV cbVG-high infection.** SG (G3BP1, magenta) and viral protein (RSV F, yellow) detection in A549 cells 24 hpi with RSV cbVG-high virus at MOI 1.5 $TCID_{50}$/cell. Zoomed in images of SG-positive cells are shown on the right with merge and RSV F single channel. Widefield image was acquired with the Apotome 2.0 at 63× magnification, scale bar = 50 μm. Measurements of CTCF are shown on the right. Data points represent the average of approximately 50 cells per group. Numerical values plotted can be found in the Supporting information: S1 Data. cbVG, copy-back viral genome; CTCF, corrected total cell fluorescence; G3BP1, GTPase-activating protein-binding protein 1; hpi, hours postinfection; MOI, multiplicity of infection; RSV, respiratory syncytial virus; SG, stress granule.
(TIF)

**S5 Fig. SG formation is not necessary for translation inhibition during SeV cbVG-high infection.** G3BP1 (green) for SG detection and PMY (red) for translation in A549 control and G3BP1/2 dKO cells infected with SeV cbVG-high (SeV NP, magenta) at MOI 3 $TCID_{50}$/cell 24 hpi. cbVG, copy-back viral genome; dKO, double KO; G3BP1, GTPase-activating protein-binding protein 1; hpi, hours postinfection; MOI, multiplicity of infection; NP, nucleoprotein; PMY, puromycin; SeV, Sendai virus; SG, stress granule.
(TIF)

**S1 Movie. SeV cbVG-dependent SG form asynchronously throughout the infection.**
G3BP1-GFP expressing A549 cells infected with rSeV-C$^{miRF670}$ reporter virus at MOI 3
TCID$_{50}$/cell with 20 HAU of supplemented cbVG purified particles, time-lapse microscopy
6–72 hpi, images every 6 h at a 20× magnification. cbVG, copy-back viral genome; HAU, hem-
agglutination unit; G3BP1, GTPase-activating protein-binding protein 1; hpi, hours postinfec-
tion; MOI, multiplicity of infection; SeV, Sendai virus; SG, stress granule.
(DOCX)

**S2 Movie. SG-positive cells show decreased virus reporter protein expression.** G3BP1-GFP
expressing A549 cells infected with rSeV-C$^{miRF670}$ reporter virus at MOI 3 TCID$_{50}$/cell with 20
HAU of supplemented cbVG purified particles, time-lapse microscopy 12–24 hpi, images
taken every 30 min at a 40× magnification using a widefield microscope. cbVG, copy-back
viral genome; G3BP1, GTPase-activating protein-binding protein 1; HAU, hemagglutination
unit; hpi, hours postinfection; MOI, multiplicity of infection; SG, stress granule.
(DOCX)

**S1 Data. All figures' data.** Excel file containing all the numerical values plotted in each graph
including all replicates, mean, standard deviation, and standard error of the mean. Each sheet
in the excel file is named after the figure the numerical values belong to.
(XLSX)

**S1 Raw Images. Blots and gels.** PDF file containing all original blots and gels used in the main
and supporting figures of the manuscript.
(PDF)

## Acknowledgments

We would like to acknowledge Dr. Susan Weiss (University of Pennsylvania) for providing the
RNAseL, PKR and MAVS KO A549 cells, Nicole Rivera-Espinal for performing the imaging
experiments for S3 Fig, and Emna Achouri for data visualization in Fig 6D.

## Author Contributions

**Conceptualization:** Lavinia J. González Aparicio, Carolina B. López.

**Data curation:** Lavinia J. González Aparicio.

**Formal analysis:** Lavinia J. González Aparicio.

**Funding acquisition:** Carolina B. López.

**Investigation:** Lavinia J. González Aparicio.

**Methodology:** Lavinia J. González Aparicio, Yanling Yang, Matthew Hackbart, Carolina B.
López.

**Resources:** Carolina B. López.

**Supervision:** Carolina B. López.

**Visualization:** Lavinia J. González Aparicio.

**Writing – original draft:** Lavinia J. González Aparicio, Carolina B. López.

**Writing – review & editing:** Lavinia J. González Aparicio, Yanling Yang, Matthew Hackbart,
Carolina B. López.

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
