## [Editor Report · Decision Letter 0]

20 Jul 2023

Dear Dr, Lopez, 

Thank you for submitting your manuscript entitled "Copy-back viral genomes induce a cellular stress response that interferes with viral protein expression without affecting antiviral immunity" for consideration as a Research Article by PLOS Biology.

Your manuscript has now been evaluated by the PLOS Biology editorial staff, as well as by an academic editor with relevant expertise, and I am writing to let you know that we would like to send your submission out for external peer review.

Once your full submission is complete, your paper will undergo a series of checks in preparation for peer review. After your manuscript has passed the checks it will be sent out for review. To provide the metadata for your submission, please Login to Editorial Manager (https://www.editorialmanager.com/pbiology) within two working days, i.e. by Jul 22 2023 11:59PM.

Kind regards,

Paula

---

Senior Editor

PLOS Biology

---

## [Decision Letter · Decision Letter 1]

14 Sep 2023

Dear Dr. Lopez,

Thank you for your patience while your manuscript "Copy-back viral genomes induce a cellular stress response that interferes with viral protein expression without affecting antiviral immunity" went through peer-review at PLOS Biology. Your manuscript and previous reviews have now been evaluated by the PLOS Biology editors, an Academic Editor with relevant expertise, and by an independent reviewer.

In light of the all the assessments and the review, which you will find at the end of this email, we are pleased to offer you the opportunity to address the comments from the reviewer in a revision that we anticipate should not take you very long. We will then assess your revised manuscript and your response to the reviewer's comments with our Academic Editor aiming to avoid further rounds of peer-review, although might need to consult with the reviewers, depending on the nature of the revisions.

Please also address the following editorial and formatting requests:

1. DATA POLICY:

A) Supplementary files (e.g., excel). Please ensure that all data files are uploaded as 'Supporting Information' and are invariably referred to (in the manuscript, figure legends, and the Description field when uploading your files) using the following format verbatim: S1 Data, S2 Data, etc. Multiple panels of a single or even several figures can be included as multiple sheets in one excel file that is saved using exactly the following convention: S1_Data.xlsx (using an underscore).

B) Deposition in a publicly available repository. Please also provide the accession code or a reviewer link so that we may view your data before publication. 

Regardless of the method selected, please ensure that you provide the individual numerical values that underlie the summary data displayed in the following figure panels as they are essential for readers to assess your analysis and to reproduce it: Figures 1BDF, 2BD, 3B, 4CE, 5B, 6ABCDFG, 8AC and Supplementary Figure S1C.

For manuscripts submitted on or after 1st July 2019, we require the original, uncropped and minimally adjusted images supporting all blot and gel results reported in an article's figures or Supporting Information files. We will require these files before a manuscript can be accepted so please prepare and upload them now. We require this for Figures 4AB, 5A, 6EH and Supplementary Figures S1B, S3AB.

Please carefully read our guidelines for how to prepare and upload this data: https://journals.plos.org/plosbiology/s/figures#loc-blot-and-gel-reporting-requirements

3. Please provide a blurb which (if accepted) will be included in our weekly and monthly Electronic Table of Contents, sent out to readers of PLOS Biology, and may be used to promote your article in social media. The blurb should be about 30-40 words long and is subject to editorial changes. It should, without exaggeration, entice people to read your manuscript. It should not be redundant with the title and should not contain acronyms or abbreviations.

**IMPORTANT - SUBMITTING YOUR REVISION**

*Resubmission Checklist*

*Published Peer Review*

*PLOS Data Policy*

*Blot and Gel Data Policy*

Sincerely,

Paula

---

Senior Editor

PLOS Biology

REVIEWS:

Reviewer #1: Arbitrating reviewer. Virus-host ribonucleoprotein complexes and innate immunity.

Reviewer #1: The manuscript by Gonzalez Aparicio entitled "Copy-back viral genomes induce a cellular stress response that interferes with viral protein expression without affecting antiviral immunity" links 5' copy-back DVG production upon infections with Sendai or respiratory syncytial virus (RSV) to PKR-dependent formation of canonical SGs complemented by the down-regulation of viral translation. This 5'-copy-back DVG functioning is independent 5'-copy-back functioning in activation of IFN signaling. Presented results are novel and important to understand various mechanisms of virus-host interactions, providing an example where the same pathogen molecular pattern is implemented in several cellular pathways.

However, I have several major remarks:

1) Pfaller et al. J.Virology 2014 paper (PMID: 24155404) should be discussed and cited in the current manuscript. As earlier, using measles virus (another Paramyxoviridae), Pfaller et al. have demonstrated that the deltaC measles virus strain produces dsRNA (copy-back DVG), activates PKR which correlates with SG formation. Thus Plaller et al. study provides initial information on specific RNA molecules -copy-back DVG that trigger PKR signaling and stress granule formation during natural viral infections.

2) The abbreviation of copy-back viral genomes as cbVGs is perturbing. The authors describe 5'-copyback defective viral genomes that are classically abbreviated as copy-back DVG (cbDVGs, by the same and other laboratories: PMID: 30245734, PMID: 26443454, PMID: 29861183, PMID: 34160256, PMID: 34815573 etc) or as copy-back DI RNAs (cb DI-RNA: PMID: 7037195, PMID: 16631220, PMID: 35632848, PMID: 28768856, PMID: 24155404). I believe one of these already established abbreviations should be used throughout the manuscript. Idem for standard genomes (stVG-high) which is simply a full-length genome and used only 3 times in the text of the manuscript. Non-standard viral genomes (nsVGs) are DVG and why they are "hypermutated RNAs": cb DVGs are not hypermutated they just have 5'/3' complementary ends. But their sequences are stable. I addition, Paramyxoviridae respect the so called "rule of six" (PMID: 8392616).

3) IF microscopy images for all figure (except supplementary figures) are difficult to view. Other main figures, like Figure 6 are also unreadable. Figure with a better resolution should be provided. For many central (Figs. 3C, 3D, 5C, 5D, 7, and supplementary S2, S4, S5) figures quantification analyses (ex. % infected SG positive cells) should be included. I believe that the information on % non-infected SG positive cells is also important to add.

4) Western blots should contain information on number of biological replicates and quantification should be added when need (Figs 4A, 6H, S3).

5) In the abstract part, more careful communication of the observed results with SG formation and antiviral response should be considered as "SG positive cells are only 10 to 20% of the infected population at a given time,.." and " Although we cannot rule out that a reduction on ISG protein expression occurs specifically in SG positive cells, this potential reduction does not affect the global immune antiviral response during the infection,..". 

Minor

1) Additional read out "as a proxy for virus replication" (using another viral RNA or intergenic region could be considered validating the results presented on Figures 4C, 5B, and S1C.

2) Short title on the first page is : "Stress responses during pneumovirus and paramyxovirus infections" and on line 13: "Stress responses during RNA virus infections". Which one is correct?

3) For Fig/8 the figure legend is missing, and the Fig.8D is unreadable!

4) Several titles in materials and methods are written in blue.

5) Line 563 S3 or S1? 

6) On line 383 (E) should be (H). I would suggest to add additional replicates to Figure 6F.

---

## [Editor Report · Decision Letter 2]

12 Oct 2023

Dear Dr Lopez,

Thank you for your patience while we considered your revised manuscript "Copy-back viral genomes induce a cellular stress response that interferes with viral protein expression without affecting antiviral immunity" for publication as a Research Article at PLOS Biology. This revised version of your manuscript has been evaluated by the PLOS Biology editors, and the Academic Editor.

Based on our Academic Editor's assessment of your revision, we are likely to accept this manuscript for publication, provided you satisfactorily address the following data and other policy-related requests.

1. DATA POLICY:

A) Supplementary files (e.g., excel). Please ensure that all data files are uploaded as 'Supporting Information' and are invariably referred to (in the manuscript, figure legends, and the Description field when uploading your files) using the following format verbatim: S1 Data, S2 Data, etc. Multiple panels of a single or even several figures can be included as multiple sheets in one excel file that is saved using exactly the following convention: S1_Data.xlsx (using an underscore).

B) Deposition in a publicly available repository. Please also provide the accession code or a reviewer link so that we may view your data before publication.

Regardless of the method selected, please ensure that you provide the individual numerical values that underlie the summary data displayed in the following figure panels as they are essential for readers to assess your analysis and to reproduce it: We are missing this for **Figures 6D and S1C.**

We require the original, uncropped and minimally adjusted images supporting all blot and gel results reported in an article's figures or Supporting Information files. We will require these files before a manuscript can be accepted so please prepare and upload them now. We are missing this for **Figure S1B**.

Please carefully read our guidelines for how to prepare and upload this data: https://journals.plos.org/plosbiology/s/figures#loc-blot-and-gel-reporting-requirements

We expect to receive your revised manuscript within two weeks.

*Published Peer Review History*

*Press*

Sincerely,

Paula

---

Senior Editor,

pjaureguionieva@plos.org,

PLOS Biology

---

## [Editor Report · Decision Letter 3]

15 Oct 2023

Dear Dr Lopez,

Thank you for the submission of your revised Research Article "Copy-back viral genomes induce a cellular stress response that interferes with viral protein expression without affecting antiviral immunity" for publication in PLOS Biology. On behalf of my colleagues and the Academic Editor, Andrew Mehle, I am pleased to say that we can in principle accept your manuscript for publication, provided you address any remaining formatting and reporting issues. These will be detailed in an email you should receive within 2-3 business days from our colleagues in the journal operations team; no action is required from you until then. Please note that we will not be able to formally accept your manuscript and schedule it for publication until you have completed any requested changes.

PRESS

Sincerely, 

Paula

---

Senior Editor

PLOS Biology
